# Global Climate System Response to SOFIA Antarctic Meltwater in HadCM3-M2.1

Amar Mistry[1], Dan Lunt[1], and Xin Ren[1]

[1]School of Geographical Sciences, University of Bristol, University Road, Bristol BS8 1SS, UK

**Correspondence:** Amar Mistry (amar.mistry.2018@bristol.ac.uk)

**Abstract.** As the climate continues to warm, the Antarctic Ice Sheet (AIS) and its surrounding floating ice shelves are becoming increasingly susceptible to rapid collapse. Despite the potential impact this poses to the global climate system, the effects of ice-climate feedbacks are not directly considered by most existing coupled climate models, including those in the most recent Coupled Model Intercomparison Project (CMIP6). As such, there remains much uncertainty over the impact of this additional meltwater on current global climate change projections. Here, we use the coupled atmospheric-ocean general circulation model HadCM3-M2.1 to study the effect of a continuous meltwater discharge from the AIS on the global climate system. This involves carrying out a series of freshwater hosing experiments based on the newly proposed Southern Ocean Freshwater Input from Antarctica (SOFIA) Initiative. Due to the relative computational efficiency of the HadCM3-M2.1 model, we are able to explore longer timescales than is usual. We find that $\sim$1000 years of continuous meltwater drives global atmospheric cooling, sea ice expansion in both hemispheres and a northward shift of the Intertropical Convergence Zone (ITCZ). The resulting freshening of the global ocean results in the weakening of both the AMOC and AABW. This triggers pervasive ocean warming at depths greater than 5000m. An additional sensitivity study is also conducted in which the sensitivity of the climate model response to a change in the horizontal distribution of AIS meltwater is tested. As a result, we find that the manner in which the AIS loses mass, whether that be predominately through iceberg mass loss or basal melt, is unlikely to affect the global climate response to AIS meltwater.

## 1 Introduction

The Antarctic Ice Sheet (AIS) is an area of increasing interest as the largest potential contributor to global sea level rise (Meredith et al., 2019; Noble et al., 2020). Between 1992-2020, the AIS lost 2670 [1800 to 3540] Gt mass, contributing 7.4 [5.0 to 9.8] mm to global mean sea levels (Fox-Kemper et al., 2023). However, as atmospheric and ocean temperatures continue to warm under anthropogenic-induced climate change, the rate of mass loss from the AIS is expected to increase. The warming and non-iceshelf component of the mass loss is expected to result in AIS meltwater into the Southern Ocean (SO) becoming a primary contributor to global sea level rise in the coming decades (Fox-Kemper et al., 2023). Current projections suggest that the AIS will likely contribute 30 to 340 mm under SSP5-8.5 by the end of the coming century (Fox-Kemper et al., 2023).

Meltwater from the AIS not only affects global sea levels, but also has a direct influence on the SO (Fox-Kemper et al., 2023; Swart et al., 2023). As the largest relative contributor to the oceanic sink of anthropogenic heat and carbon, the SO is important

in mediating climate change (Meredith et al., 2019; Swart et al., 2023). Previous studies have investigated the global climate system response to AIS meltwater through the use of freshwater hosing experiments, in which additional freshwater is imposed in a coupled climate model (Table S2) (Swart et al., 2023). These studies found that freshwater input from the AIS can cause significant freshening and cooling of sea surface temperatures (SSTs) across large parts of the SO and global oceans, as well as widespread subsurface warming and changes to global precipitation patterns (Golledge et al., 2019; Purich and England, 2023; Sadai et al., 2020; Li et al., 2023b; Bronselaer et al., 2018).

However, despite this, ice-climate feedbacks have generally been excluded from coupled climate model simulations, including those in the most recent Coupled Model Intercomparison Project (CMIP6) (Swart et al., 2023; Eyring et al., 2016). In addition, previous studies have not been evaluated against each other and, whilst there is a general agreement between different model responses to additional freshening in the SO, there remains much uncertainty over the magnitude of this response (Swart et al., 2023). This lack of evaluation and associated uncertainty is in part due to an inconsistency in experimental design across the literature (Swart et al., 2023). Experimental design varies across a number of different factors, including; amount of freshwater flux added to the model, the rate and duration of this flux forcing, and the location it is added (Table S2) (Swart et al., 2023).

As such it is important to assess the sensitivity of the global climate system to a change in the horizontal distribution of AIS meltwater. Using a fully coupled earth system model, Pauling (2016) found that the sea ice response is insensitive to a change in horizontal distribution. However, to date, there have been few studies that investigate the effect of a change in horizontal distribution of AIS meltwater on the global climate response. As such, there remains much uncertainty surrounding this issue, and further work exploring the global climate system response to a range of distributions over longer time scales is an important next step.

It is also important to note that, to date, there have been few studies that project future climate responses to freshwater flux from the AIS beyond a few hundred years (Table S2), despite the fact that current warming has the potential to result in anomolous meltwater fluxes from Antarctica for at least a millenium (Fox-Kemper et al., 2023). Seidov, Barron and Haupt (2001) conducted a series of experiments using an ocean-only model to investigate the climate response to freshwater flux from the AIS over a period of 2000 years, whilst Menviel et al. (2010) ran a model simulation for 800 model years (WAIS 800) using a coupled climate model of intermediate complexity (EMIC). Both studies found that AIS meltwater had significant long term climatic impacts; notably, Seidov, Barron and Haupt (2001) found that freshening of the SO led to significant changes to the global ocean's circulation, resulting in the SO becoming the major climatic player in long-term climate change. However, it is unclear whether results from simplified EMICs and ocean-only models would hold in fully coupled AOGCMs, as the simplified models do not fully account important atmospheric or cryospheric feedbacks, and are therefore limited in their ability to assess the climate response to AIS meltwater (Seidov et al., 2001). Most recently, Fogwill et al. (2015) carried out the longest projection using a fully coupled climate model to date, at 1000 years into the future. However, Fogwill et al. (2015) focuses primarily on results obtained from a relatively short experiment (200 years), and the experiments conducted only consider freshwater input from the WAIS. This lack of research on long-term effects suggests a clear gap in our understanding of the long-term, millennial scale climate response to freshwater input from the AIS. Exploring long-term impacts of AIS

meltwater will allow us to investigate potential global and AIS based climate feedbacks. In addition, it would also allow us to further develop our understanding of paleo-climates because it is likely that the AIS experienced significant meltwater episodes on multi-millennial scales in the past, which contributed to considerable changes to the Earth's climate (Noble et al., 2020; Bentley et al., 2014; Stuhne and Peltier, 2015; Golledge et al., 2014).

This paper aims to explore the long term impacts of a rapid collapse of the AIS on the global climate system through a series of freshwater hosing experiments that are designed to mimic the effects of global ice-climate feedbacks. The experimental design is taken from the Southern Ocean Freshwater Input from Antarctica (SOFIA) Initiative, a recently proposed project that aims to measure the climate system response to AIS meltwater by coordinating a multi-model ensemble of simulations using a common experimental design (Swart et al., 2023). As mentioned previously, inconsistent experimental design across

the existing literature is creating uncertainties over the magnitude of the global climate system's response to AIS meltwater, inhibiting our ability to constrain the climate impacts of AIS meltwater. By drawing upon and contributing to the SOFIA Initiative, this study aims to improve our understanding of the global climate response to AIS meltwater through the use of a common experimental design.

This paper is structured as followed; Section 2 consists of a description of the climate model and experimental design; in

Section 3 the modelled response to AIS meltwater will be discussed, with a focus on climatically important parameters; and finally all conclusions will be summarised in Section 4.

## 2    Methodology

### 2.1    Model Description

In order to assess the long term climate response to AIS meltwater, an efficient and fully coupled model is needed. This

study uses a variation of HadCM3, HadCM3-M2.1, a three-dimensional fully dynamic, coupled atmospheric-ocean general circulation model (AOGCM) without flux adjustment. The model was produced by the UK Hadley Centre/Met Office and has since been updated and further developed by the Bristol Research Initiative for the Dynamic Global Environment (BRIDGE) (Valdes et al., 2017). HadCM3 has been extensively used for scientific studies of paleo (e.g. Williams et al. (2021); Kennedy-Asser et al. (2020); Farnsworth et al. (2021)) and future climate (e.g. Johns et al. (2003); Smith et al. (2007); Stott et al.

(2001)), and has also been used in the third, fourth and fifth IPCC reports (Irvine et al., 2013; Fox-Kemper et al., 2023). HadCM3 is well known for its ability to perform well with relatively fast computational speed, making it particularly valuable for running long-term simulations (Collins et al., 2001; Valdes et al., 2017). HadCM3-M2.1 incorporates an upgraded land surface scheme, MOSES2.1, as a means of incorporating a dynamic vegetation module (Valdes et al., 2017). This dynamic vegetation module is represented with nine surface types; five plant functional types, soil (desert), urban, water (lake) and

ice. MOSES2.1 also has major upgrades to all aspects of land surface exchange and surface radiation scheme compared to its predecessor, MOSES1 (Valdes et al., 2017). The atmospheric component of HadCM3 has a Cartesian grid with a horizontal resolution of 3.75°longitude x 2.5°latitude with 19 vertical layers, and uses a 30 minute time step (Valdes et al., 2017). A number of parameterisation schemes are required to represent sub-gridscale atmospheric processes, including cloud formation

and convection (Gregory et al., 1997; Valdes et al., 2017). The ocean component has a horizontal resolution of 1.25°x 1.25° with 20 vertical layers, and uses a 1 hour time step (Gordon et al., 2000; Pope et al., 2000; Valdes et al., 2017). The ocean component is based on the model of Cox (1984); it is a full primitive equation, three-dimensional model of the ocean (Valdes et al., 2017). The model uses a rigid lid which eliminates fast external mode gravity waves found in the real ocean, allowing for longer time steps and results in no variation in the volume of the ocean. The ocean component requires a number of sub-gridscale parameterisations, including momentum and horizontal eddy mixing (Valdes et al., 2017). Momentum mixing is approximated using diffusion that is controlled by a coefficient consisting of a constant background value and a term dependent on the local Richardson number (Valdes et al., 2017). Horizonal mixing of momentum is also parameterised using a latitudinally varying formulation (Valdes et al., 2017). This parameterisation, coupled with a finer resolution of the ocean grid, enables western boundary currents to be resolved (Valdes et al., 2017). Horizontal eddy mixing of tracers is parameterised using the isopycnal parameterisation of Gent and McWilliams (1990), with thickness diffusion coefficients modified using the methods described by Visbeck et al. (1997). It is important to note that, although the ocean resolution here is similar to most CMIP6 models, HadCM3-M2.1 is not eddy resolving. As noted by Frigola et al. (2025), overturning and barotropic circulations of the North Atlantic are significantly improved in eddy-resolving models. As such, this parameterisation may affect model results, including the relative strength of the Atlantic Meridional Overturning Circulation (AMOC) (Frigola et al., 2025).

Sea ice is represented as a zero layer model on top of the ocean grid. Sea ice dynamics, including ice thickness and concentration, are simply parameterised; surface wind stress over sea ice is instead applied to the ocean beneath it, with ice and accumulated snow drifting following the ocean currents in the top model layer (Gordon et al., 2000). Sea ice is assumed to have a constant salinity, with salt/freshwater fluxes formed by sea ice formation/melting represented as virtual salinity fluxes into the ocean (Valdes et al., 2017). Ice sheets in the model are not modelled dynamically (Valdes et al., 2017). As such, snow cover is allowed to build up indefinitely over land grid cells. This build up is balanced by loss through a notional iceberg calving that is represented as a time-invariant freshwater flux (Valdes et al., 2017; Ross, 2023). This flux is converted to a virtual salinity flux due to the use of a rigid lid (Fig. S1), and is distributed into polar oceans (Valdes et al., 2017). As HadCM3-M2.1 uses a rigid-lid, the ocean surface is not allowed to deform and, as a result, total ocean volume remains constant. As such, any freshwater fluxes simulated in the model do not physically increase total ocean volume, but are instead applied as virtual salinity fluxes that reduce ocean surface salinity. These virtual salinity fluxes effectively remove salt mass from the model ocean, mimicking the dilution effect that would occur with a physical ocean volume increase.

In all simulations, we are using a version of the model that has been tuned following the approach outlined by Ren et al. (2025) to give polar amplification that is in better agreement with geological proxy evidence from past climates, while not degrading the modern climate (Ren et al., 2025). Table S3 describes the parameter tuning implemented, including a brief description of the impacts.

## 2.2 Experimental Design

As summarised in Table 1, this paper conducts a series of freshwater hosing experiments from the SOFIA experimental design (Swart et al., 2023). These experiments differ from previous studies using freshwater hosing experiments (e.g. Sadai et al.

(2020); Bronselaer et al. (2018); Golledge et al. (2019)) as this study imposes a constant salinity flux as a proxy for meltwater, and as such, this meltwater flux is not derived from any ice sheet model.

A control experiment (*piControl*) with no additional AIS freshwater anomaly is conducted and run for a total of 2150 model years. As discussed previously, the use of a virtual salinity flux to account for the indefinite build-up of snow cover on land grid cells means that *piControl* will also be subject to a constant and uniform freshwater flux of $\sim 0.125$ Sv (Sverdrup; $1\text{Sv} = 1 \times 10^6 \text{m}^3\text{s}^{-1}$). The freshwater hosing (FW) experiments (*antwater* and *antwater 60S* experiments) then branch from the *piControl* experiment 1000 model years in and run for 1150 model years (Fig. 2). This initial 1000 model year 'spin up' ensures

that the climate model is calibrated to pre-industrial conditions before any additional freshwater flux is added. The differences between the *piControl* and FW experiments are taken as modelling responses. This results in a total of three simulations (Table 1).

In the *antwater 60S* experiment, a freshwater flux is released uniformly south of 60°S (Fig. 1). This horizontal distribution has also been used previously to measure the effect that northward distribution by icebergs might have on the climate (IMBIE

team, 2018; Rignot et al., 2019; Swart et al., 2023), and is commonly used to model a broader distribution of freshwater (Stouffer et al., 2007; Ma et al., 2013; Purich and England, 2023). An additional experiment (*antwater*) is also run in which meltwater is added uniformly around the Antarctic continent in the two grid cells against the coast (corresponding to 2.5° latitude away from the coastline) (Table 1; Fig. 1). Changing the horizontal distribution allows us to determine whether the climate response to AIS freshwater differs if meltwater is released from iceberg melt (*antwater 60S*) or ice shelf basal melt

exclusively (*antwater*). For both distributions, a freshwater flux is added at the surface of the ocean, which when area integrated represents a constant and uniform flux of 0.1 Sv. In both *antwater* and *antwater60S*, the freshwater flux imposed into the model is in addition to the baseline freshwater flux simulated in the *piControl* experiment, ensuring that the experiments are consistent with the SOFIA initiative's 'antwater 60S' and 'antwater' experiment designs (Table S1) (Swart et al., 2023).

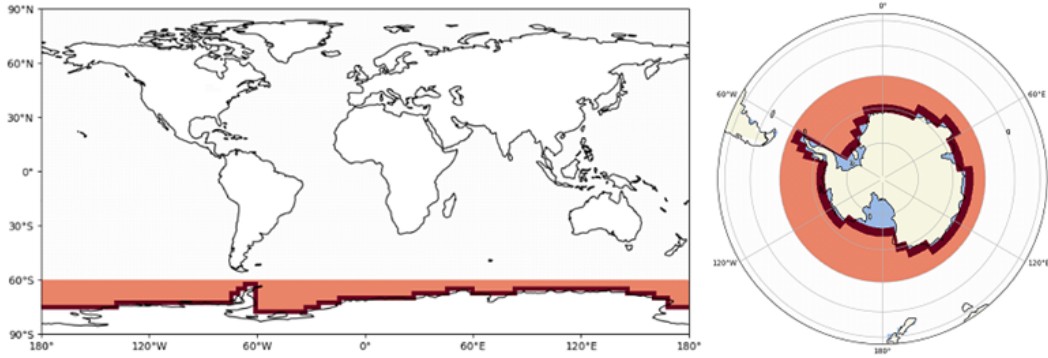

**Figure 1.** The horizontal distribution of freshwater inputs to background flux. The area shaded dark red refers to the AIS adjacent distribution (*antwater* experiment), the area shaded orange shows the distribution south of 60°S (*antwater 60S* experiment). The area shaded light blue shows ice shelves surrounding the AIS.

**Table 1.** Experiment name, the amount of freshwater forcing in Sv, description of how the freshwater forcing is applied, length of experiment in model years, and other external forcings.

| Name (PUMA name) | Freshwater Perturbation (Sv) | Distribution | Time span (years) | Additional Forcings |
|---|---|---|---|---|
| *piControl* (xpurb) | Background | N/A | 2150 | Fixed preindustrial |
| *antwater 60S* (xpurd) | Background + 0.1 Sv | South of 60°S | 1150 | Fixed preindustrial |
| *antwater* (xpzdb) | Background + 0.1 Sv | AIS adjacent | 1150 | Fixed preindustrial |

Two-way student t-tests are performed to indicate statistical significance at the 95% confidence level. Although the climate data presented here is not normally distributed, a student t-test is still considered useful in showing a broad indication of statistical significance (Rasch and Tiku, 1984).

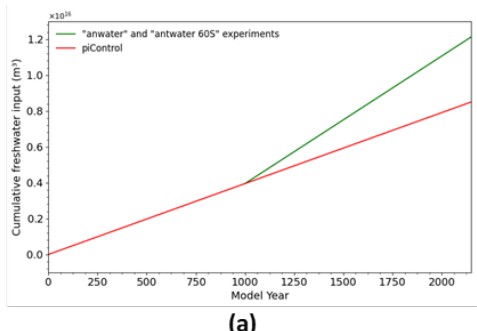
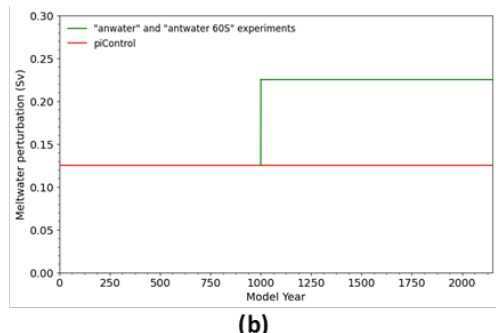

**Figure 2.** (a) Cumulative global freshwater input over 2150 model years for *piControl* (red) and FW experiments (green). (b) Meltwater time series over 2150 model years for the *piControl* and FW experiments.

Whilst the initial perturbation of 0.1 Sv far exceeds current observations of meltwater from the AIS of 148 [94 to 202] Gt yr⁻¹ (~0.005 [0.003 to 0.006] Sv), it is within range of future forcing magnitudes expected in the 21st century (Swart et al., 2023), and well within the range of previous studies (Table S1). In addition, a greater forcing is more likely to elicit a clear response from the Earth's climate system (Swart et al., 2023), allowing us to explore the climate response to meltwater perturbations from the AIS over extremely long time scales.

The global ocean salinity has a trend of +0.0003 psu per century in the *piControl* experiment, and -0.0034 psu per century in the FW experiments, with the difference being equivalent to the rate of fresh water input (Fig. S1). The positive salinity trend simulated in the *piControl* experiment can be explained by the residual of the virtual salinity flux discussed previously, and by the fact that the background flux applied in the HadCM3-M2.1 model is not quite enough to completely balance the build-up of snow on ice sheets simulated in the model, as the background flux is a fixed field, rather than being prognostically calculated (Valdes et al., 2017).

## 3 Results

### 3.1 Salinity, temperature and sea ice response

Freshening of the ocean surface is simulated throughout all ocean basins (Fig. 3). By the end of the 1150 years, mean global sea surface salinity (SSS) decreases by $\sim$ 0.15 psu (Fig. 3a), whilst mean SSS averaged over the SO (-60°S to -90°S, 0° to 360°) decreases by $\sim$ 0.125 psu (Fig. 3b). The area immediately surrounding the AIS experiences a decrease in SSS of up to 0.5 psu (Fig. 3h). Freshening around the AIS is relatively uniform, with no particular region experiencing significantly more freshening (Fig. 3f-h), and is a direct result of AIS meltwater being applied directly into this region. Whilst an initial increase

in SSS is simulated in the tropics (e.g. Southern Atlantic Ocean and Pacific Ocean) and Arctic Ocean, the majority of these areas experience freshening by the end of the simulation (Fig. 3c-e). This initial increase in SSS is not enough to offset the general trend of freshening in the Northern Hemisphere, although the Northern Hemisphere freshens at a lower rate than the Southern Hemisphere (Fig. S2).

The surface freshening of $\sim$ 0.15 psu over 1150 years described above is transported into the deep ocean over time (Fig.

4a, 4b). At the moment of meltwater input, maximum freshening of 0.04-0.05 psu is simulated in the upper 500m. By the end of the simulation, maximum freshening of 0.1-0.12 psu is simulated in the upper 500m (Fig. 4a, 4b). After 500 years of continuous meltwater input, freshening is simulated throughout the entire water column across the global oceans. A similar pattern is simulated in the SO, although freshening in the SO is more concentrated at the near surface and rapidly diminishes with depth (Fig. 4d). Additionally, whilst freshening is also simulated throughout the entire water column in the SO, this is not

simulated until after 900 years of continuous meltwater input.

The pattern and magnitude of freshening in *antwater 60S* largely matches that found in the existing literature. Here, the magnitude of freshening in the SO is relatively low, and is more consistent with average SO anomalous surface freshening simulated in Purich and England's (2023) low meltwater experiment ('ANT-MW-LOW'; -0.15 psu by the end of the century) and Li, Marshall, et al. (2023b) mid-meltwater experiment (2000 Gt/year; -0.11 psu after 50 model years). However, freshening

is not confined to the Southern Hemisphere in *antwater 60S*, and as such the spatial distribution of freshening is more closely matched to high meltwater experiments, such as 'ANT-MW-HIGH' and 'RCP8.5FW' run by Purich and England (2023) and Sadai et al. (2020) respectively. On longer time scales, Menviel et al. (2010) reported that SO SSS decreases linearly by 0.2 psu after 800 years. Recent studies have also reported that freshening simulated at the ocean surface spreads pervasively into the deep ocean; Sadai et al. (2020) and Li, England et al. (2023a) simulated freshening at depths of up to 4000m and 5000m

respectively throughout all ocean basins. In addition, Li, England et al. (2023a) 50 year simulations in which 0.08 Sv of AIS meltwater is imposed under a RCP8.5 scenario reported surface freshening in the SO of up to 0.04 psu and an increase in salinity of 0.03 psu at depths between 1000-4000m. These findings are in line with simulations from the same time period shown here, although Li, England et al. (2023a) findings are likely influenced by additional climate forcings.

Cooling of surface air temperatures (SATs) is seen in response to freshening of the global ocean surface. After an initial

period of 100 years, annual mean global SATs stabilise at a cooler equilibrium ($\sim$0.2°C cooler) (Fig. 5a). In the SO, SATs follow a similar trend; after 50 years, mean SO SATs cool by 1°C (Fig. 5b). This anomalous cooling is initially concentrated in

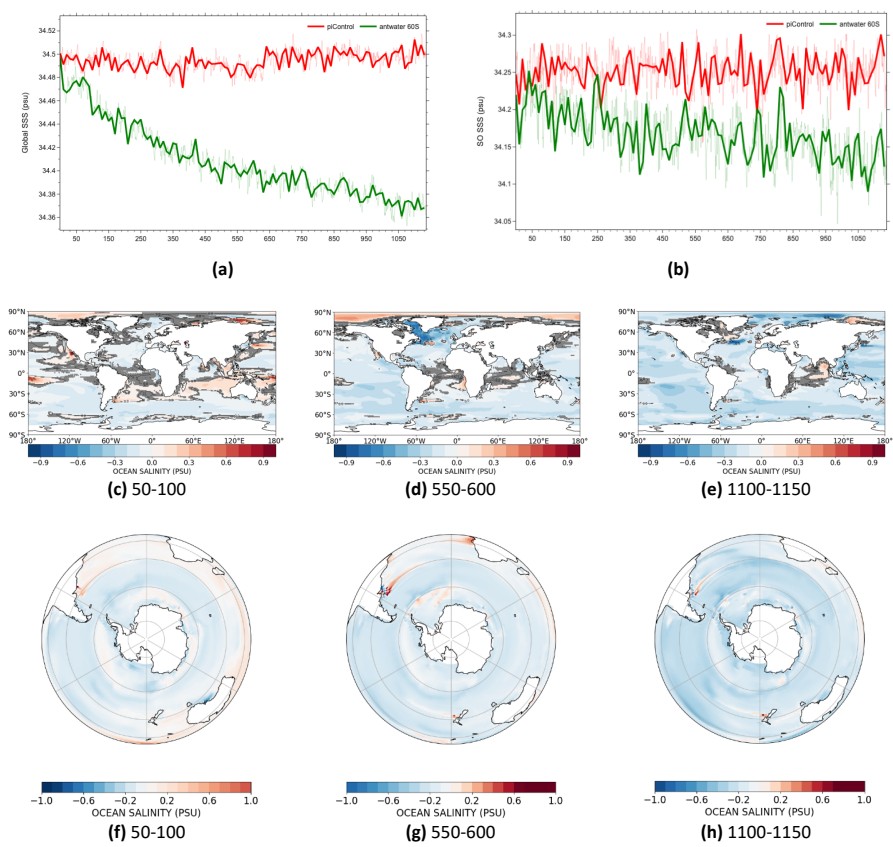

**Figure 3.** (a) Annual mean global SSS time series. (b) Annual mean SO SSS time series. *piControl* and *antwater 60S* are shown in red and green respectively. Solid lines show results smoothed with a 10-yr running average. (c) 50-100, (d) 550-600 and (e) 1100-1150 global meltwater-induced anomalies in SSS (psu) for *antwater 60S* relative to *piControl*. (f-h) Same as (c-e) for the SO. Grey area indicates less than 95% significance

the SO and the area immediately surrounding the AIS (Fig. 5c-e), with parts of the Weddell Sea and Amundsen sea region experiencing cooling of up to 3°C (Fig. 5f-h). Despite the overall global cooling of SATs, there are several regions, predominately in the Northern Hemisphere, that experience anomalous warming. Anomalous warming is simulated across the North Pacific throughout the 1150 years, whilst significant warming of up to 2°C is also simulated in Baffin Bay in 1100-1150. However, these areas of warming are not significant enough to offset the general trend of cooling simulated in the Northern Hemisphere in *antwater 60S* (Fig. S3).

Cooling of mean global and SO SATs is commonly reported as a response to AIS meltwater (Golledge et al., 2019; Li et al., 2023b; Purich and England, 2023). Whilst the magnitude of cooling simulated here is relatively low compared to many studies (e.g. Sadai et al. (2020); Purich and England (2023); Golledge et al. (2019)), this is to be expected as the magnitude of meltwater

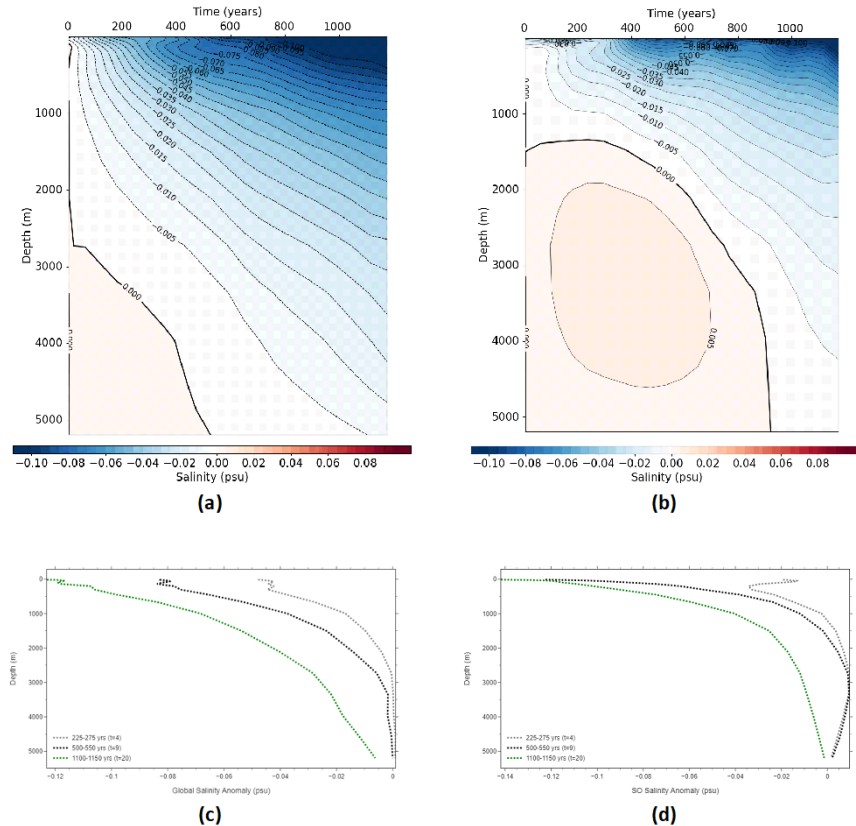

**Figure 4.** (a) Hovmuller plot (depth (m) vs time) of the anomalous global ocean salinity (psu) for *antwater 60S* relative to *piControl*. (b) Hovmuller plot of the anomalous SO salinity (psu) for *antwater 60S* relative to *piControl*. Solid black line represents the point at which the salinity anomaly changes from positive to negative. (c) Vertical profiles of anomalous global ocean salinity (psu) at different depths. Salinity profiles are calculated for periods 225-275, 500-550 and 1100-1150 (grey, black and green, respectively) for *antwater 60S* relative to *piControl*. (d) Same as (c) for SO salinity.

input in this study is also relatively small, no additional forcings are included and this study only simulates AIS meltwater. Over long time scales, Menviel et al. (2010) reported that average SO SATs fell by ∼1°C after 800 years of continuous meltwater flux, closely matching the results produced here. Additionally, Menviel et al. (2010) also reported a similar trend in decreasing SO SATs, whereby temperatures rapidly cool and stabilise at a cooler equilibrium. However, this trend has not been reported in shorter experiments; instead, SO and global SATs are generally reported to decrease linearly over time. This unusual trend simulated here is therefore likely due to the length of the simulations, although this trend may also be a response to a relatively low meltwater forcing. Spatially, results largely match those found in Purich and England (2023), in which cooling extends into the Northern Hemisphere, but no significant cooling is reported in the northern hemisphere extratropics.

Meltwater-induced surface cooling is accompanied by an increase in annual mean global sea ice thickness, with annual mean global sea ice thickness increasing by ∼ 0.01 m (Fig. 6a). Global sea ice trends follow the same pattern as SATs, whereby an

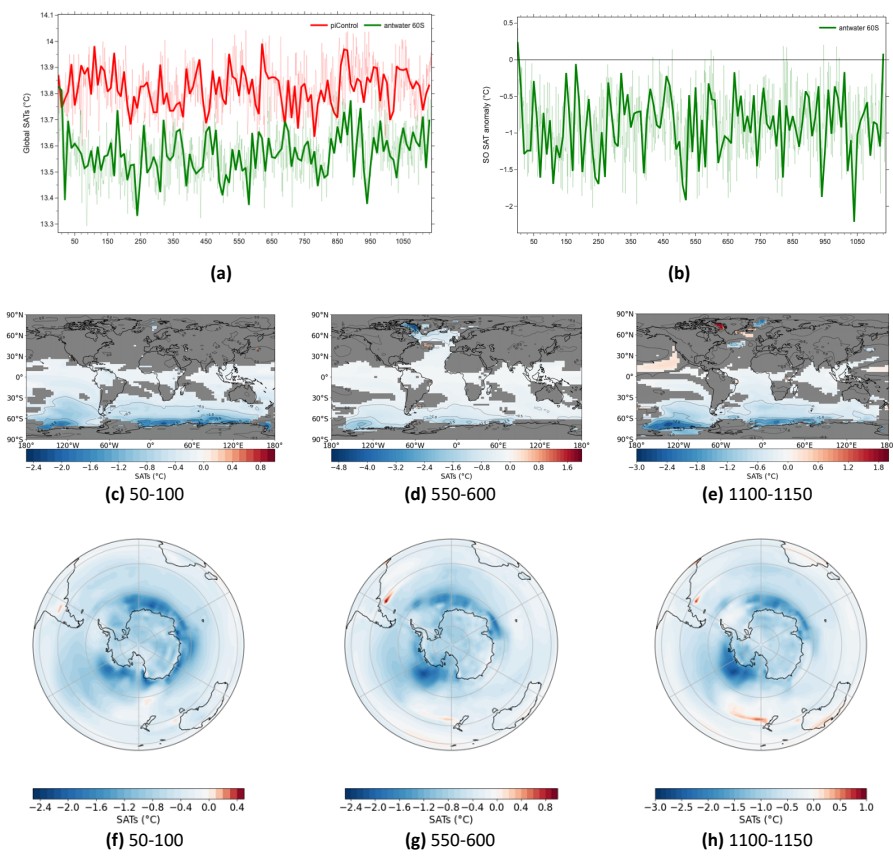

**Figure 5.** (a) Annual mean global SATs time series. (b) Annual mean SO SAT anomaly time series for *antwater 60S* relative to *piControl*. *piControl* and *antwater 60S* are shown in red and green respectively. Solid lines show results smoothed with a 10-yr running average. (c) 50-100, (d) 550-600 and (e) 1100-1150 global meltwater-induced anomalies in SATs (°C) for *antwater 60S* relative to *piControl*. (f-h) Same as (c-e) for the SO. Grey area indicates less than 95% significance.

immediate and rapid change in global mean sea ice thickness is simulated. The *antwater60S* simulation rapidly reaches a new equilibrium state, with the average global sea ice thickness between 0-100 years being $\sim 0.007$ m greater than in *piControl* (Fig. 6a). Spatially, thickening is initially simulated in both polar regions (Fig. 6e), although, on average, sea ice in the Northern Hemisphere becomes less thick over time than sea ice in the Southern Hemisphere (Fig. 6b). Off the coast of the AIS, sea ice thickens by up to 2 m, with growth concentrated around the Dronning Maud Land and Weddell Sea (Fig. 6f-h). In the Northern Hemisphere, thickening of up to 0.5m is simulated across large parts of the Arctic Ocean (Fig. 6c-e).

Sadai et al. (2020) also reported sea ice thickening across the SO, with thickening concentrated in regions in which meltwater is directly added, namely the Amundsen and Bellingshausen Sea and Ross Sea. Significant changes to sea ice in the northern hemisphere are not reported elsewhere in the literature, although Purich and England (2023) reported minimal sea ice expansion

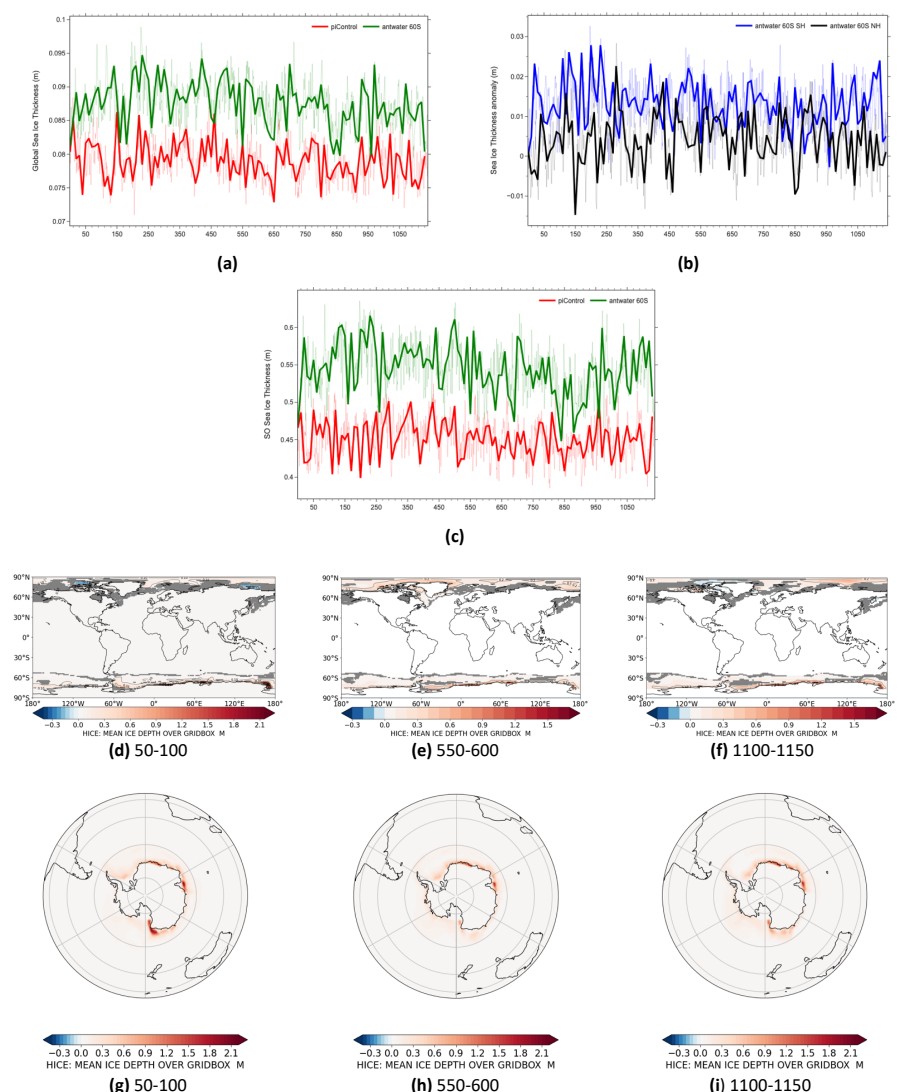

**Figure 6.** (a) Annual mean global sea ice thickness (m) time series. (b) Annual mean sea ice thickness (m) anomaly by hemisphere. NH and SH are shown in black and blue respectively. (c) Annual mean SO sea ice thickness (m) time series. *piControl* and *antwater 60S* are shown in red and green respectively. Solid lines show results smoothed with a 10-yr running average. (d) 50-100, (e) 550-600 and (f) 1100-1150 global meltwater-induced anomalies in sea ice thickness (m) for *antwater 60S* relative to *piControl*. (g-i) Same as (c-e) for the SO. Grey area indicates less than 95% significance.

of less than 10% in the Arctic Ocean in both ANT-MW-LOW and ANT-MW-HIGH experiments. Over long time scales, our findings are in line with Menviel et al. (2010) WAIS800 experiment, which also reported significant increases in sea ice extent of up to 17 x $10^{12}$ km$^2$ across the SO by the end of the 800 years.

### 3.2  Subsurface temperatures and ocean circulation response

Cooling of global SSTs is accompanied by a warming of global oceans at depth (Fig. 7a). Stratification of waters surrounding
the AIS caused by the continuous injection of freshwater leads to a reduction in convective overturning and upwelling of
subsurface waters to the surface (Purich and England, 2023). This causes heat to remain at depth. Over time, the depth at which
warming is simulated becomes more shallow (Fig. 7a and 7c). After 1150 years of continuous meltwater input, the depth at
which warming is simulated rises from 1000m to 500m (Fig. 7a). By the end of the simulation, maximum warming of up to
0.15°C is simulated at depths of 2500-3000m (Fig. 7a and 7c). A similar pattern of warming at depth is simulated in the SO,
although warming occurs at shallower depths of 400m (Fig. 7b and 7d). In addition, the degree of warming at depth is greater
in the SO, with warming of up to 0.18°C simulated at depths of 1000-3000m between 1100-1150.

At extreme depths of 2500m, annual mean global temperatures increase linearly, reaching 0.16 °C warmer than in *piControl*
at the end of the 1150 years (Fig. 8a). In the SO, temperatures warm by up to 0.15 °C within the first 500 years, after which
temperatures stabilise at this warmer state for the remainder of the 1000 year simulation, before rising again in the last 100
years (Fig. 8b). Slight cooling at depth is simulated across large areas in the Northern Hemisphere; much of the Atlantic and
Arctic Ocean experience cooling of less than 0.5 °C by the end of the simulation (Fig. 8e).

Subsurface warming simulated here is consistent with the current literature. Purich and England (2023) found that zonal-
mean subsurface warming in the SO extends from 100m to below 2000m in both ANT-MW-LOW and ANT-MW-HIGH
experiments. Li, England et al. (2023a) also reported significant subsurface warming in the SO as a response to Antarctic
meltwater. Globally, the distribution of 2500m-depth temperatures simulated here closely match Purich and England's (2023)
experiments, where subsurface warming extends northward into the north Atlantic. Additionally, the strongest warming is
simulated around the coast of the AIS (notably the Ross, Amundsen and Bellingshausen Seas), in line with findings from
Purich and England (2023) and Golledge et al. (2019). Over long time scales, Menviel et al. (2010) reported similar findings
of anomalous warming in the deep SO.

Another response to increased stratification caused by AIS meltwater is a change in the location of deep-water formation in
the HadCM3-M2.1 model. In particular, AIS meltwater results in a decrease in the mixed layer depth (MLD) in the Ross Sea
during austral wintertime, and a decrease in MLD in the North Atlantic during austral summertime (Fig. 9). This reduction in
MLD corresponds to a reduction in deep-water formation in both regions, which is likely to cause weakening of ocean currents
in these regions.

Here, the AMOC strength is defined as the maximum Atlantic overturning stream function at 33°N. Although the AMOC in
this model is relatively insensitive to AIS meltwater, a slight decline in AMOC strength is simulated; the mean AMOC strength
across the 1150 years decreases from 15.5 Sv to 14.9 Sv (-0.6 Sv) (Fig. 10a). The decline in AMOC strength simulated is sta-
tistically significant to the 99% confidence level (t = 15.02, p < 0.01). This is consistent with findings presented by Li, Marshall
et al. (2023b). As noted by Golledge et al. (2019), GIS meltwater has a different effect on AMOC to AIS meltwater. Whilst the
AMOC is far more sensitive to changes in GIS meltwater, AIS meltwater has a greater impact on the lower (anticlockwise) cell
of the AMOC, which is weaker and responds more slowly to meltwater fluxes (Golledge et al., 2019). Changes in Antarctic

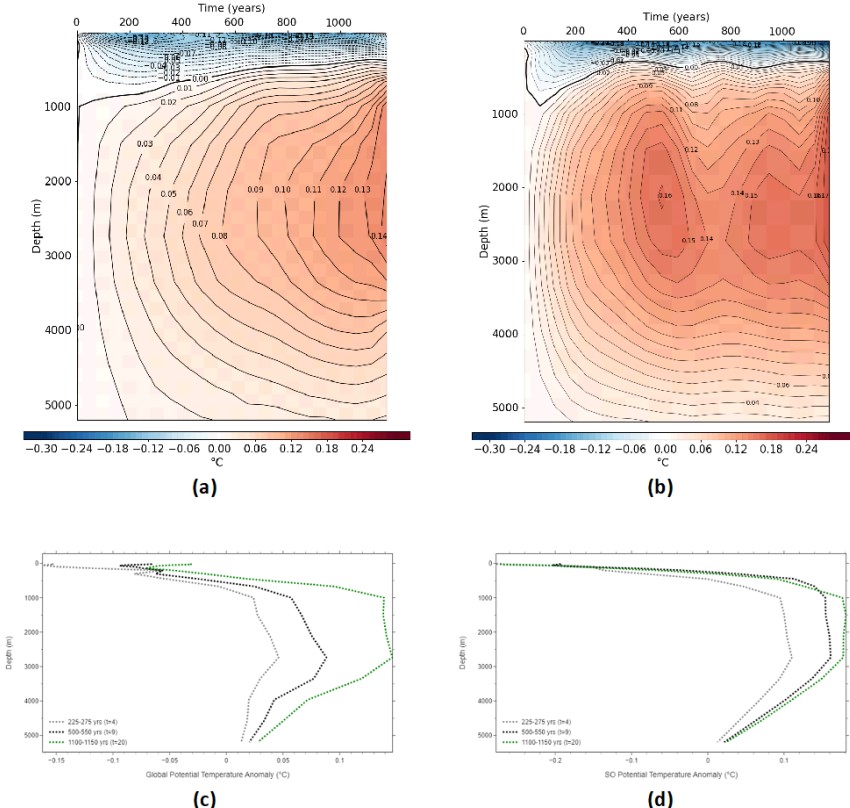

**Figure 7.** (a) Hovmuller plot (depth (m) vs time) of anomalous global ocean temperatures (°C) for *antwater 60S* relative to *piControl*. (b) Hovmuller diagram of the anomalous SO temperatures (°C) for *antwater 60S* relative to *piControl*. Solid black line represents the point at which the salinity anomaly changes from positive to negative. (c) Long-term trends of anomalous global ocean temperatures (°C) at different depths. Trends are calculated for periods 225-275, 500-550 and 1100-1150 (grey, black and green, respectively) for *antwater 60S* relative to *piControl*. (d) Same as (c) for SO temperatures.

Bottom Water (AABW) transport are also simulated. Here, the definition of AABW transport is taken from Li, Marshall et al. (2023b), and is defined as the magnitude of the minimum global-overturning stream function between 40° and 50°S, which reflects the strength of the lower cell. Unlike the AMOC, the AABW has a greater response to AIS meltwater, in line with results reported by Purich and England (2023) and Li, England et al. (2023a). The mean AABW transport anomaly across 1150 years is -2.5 Sv (Fig. 10b). The decline in AABW strength simulated is statistically significant to the 99% confidence level (t = -45.48, p < 0.01). Although there is much variability in the AABW transport anomaly, the AABW weakens almost immediately, and stabilises at a weaker state (Fig. 10b). This trend of immediate weakening and then stabilisation is also simulated by other studies investigating the climate response to AIS meltwater over long time scales (e.g. Fogwill et al. (2015); Menviel et al. (2010)).

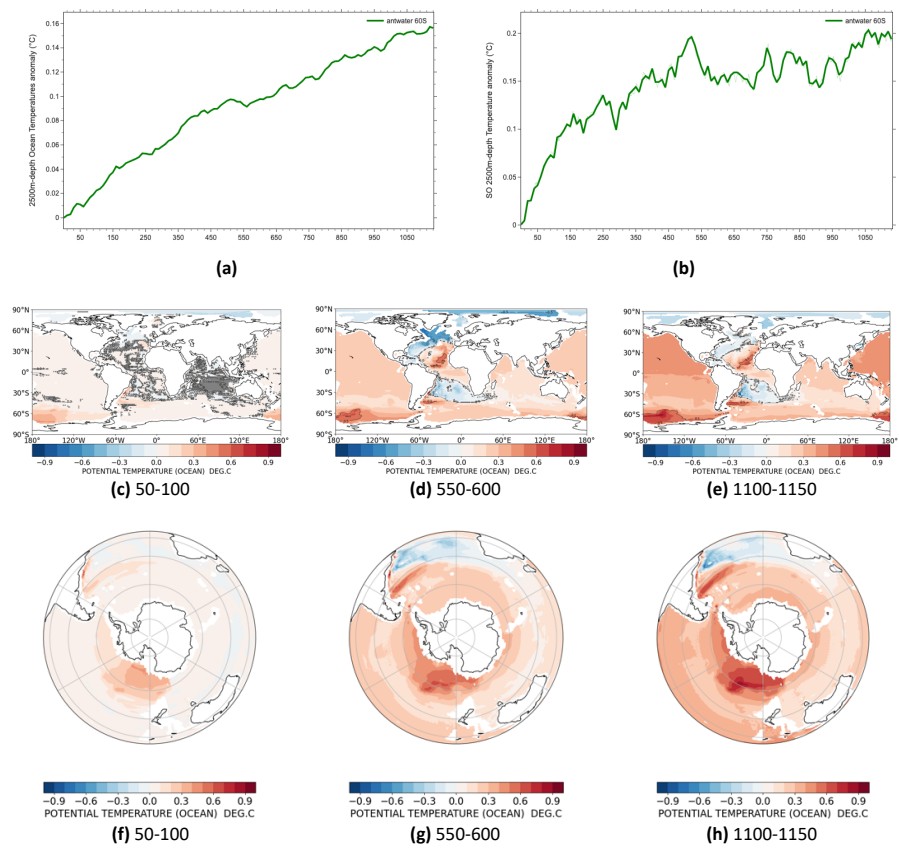

**Figure 8.** (a) Annual mean global subsurface temperatures (2500m) (°C) anomaly time series for *antwater 60S* relative to *piControl*. (b) Annual mean SO subsurface temperatures (2500m) (°C) anomaly time series for *antwater 60S* relative to *piControl*. Solid lines show results smoothed with a 10-yr running average. (c) 50-100, (d) 550-600 and (e) 1100-1150 global meltwater-induced anomalies in subsurface temperatures at 2500m (°C) for *antwater 60S* relative to *piControl*. (f-h) Same as (c-e) for the SO. Grey area indicates less than 95% significance.

Another interesting effect to consider is the impact of AIS meltwater on the Antarctic Circumpolar Current (ACC). Here, we observe changes in the volume transport through the Drake Passage (Zika et al., 2012; Park and Latif, 2019). Volume transport through the Drake Passage is calculated as the difference in barotropic stream function between the southernmost tip of South America and the northernmost tip of the Antarctic Peninsula. Here, ACC strength is initially stable during the first 100 years, after which it starts to slow and stabilises at a strength ∼10 Sv lower than in *piControl*. A gradual increase is simulated after 400 years, and ACC strength reaches the value simulated in *piControl* after 550 years (Fig. 10c). The changes in ACC simulated are statistically significant to the 99% confidence level (t = 14.95, p < 0.1). Initial weakening and partial recovery of

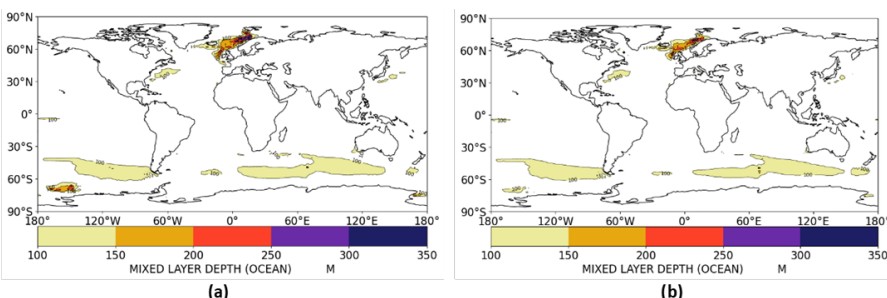

(a)                                                                              (b)

**Figure 9.** (a) Map of annual mean mixed layer depth (MLD) from *piControl* and (b) *antwater 60S* between 1100-1150. Deep water formation simulated in the northern hemisphere occurs during austral summertime, and deep water formation simulated in the southern hemisphere occurs during austral wintertime.

the ACC is also simulated in Park and Latif (2019) GWMW10 simulation, although they attribute this recovery to the presence of $CO_2$-forcing in the experiment that offsets the collapse of the SO MOC.

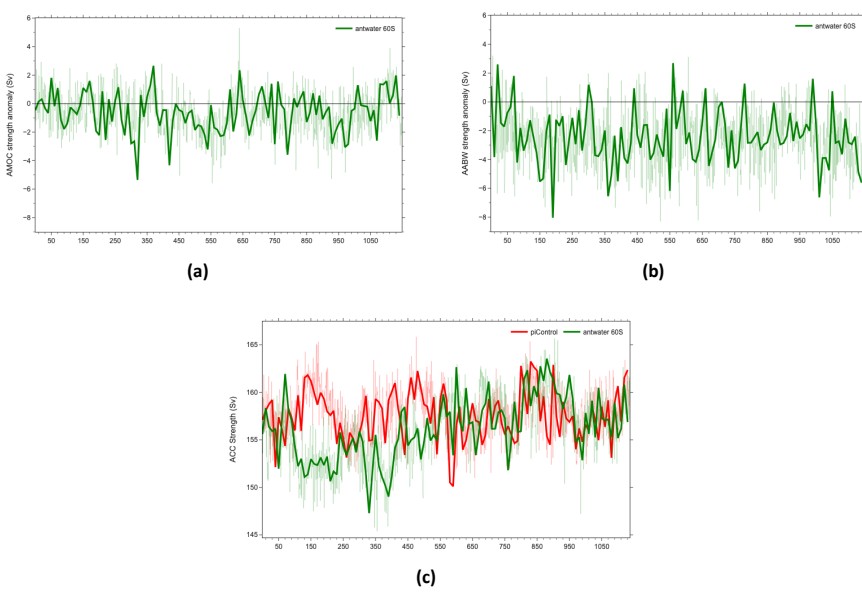

**Figure 10.** (a) Annual mean AMOC strength anomaly (Sv) and (b) Annual mean AABW strength anomaly (Sv) time series for *antwater 60S* relative to *piControl*. (c) Annual mean ACC strength (Sv) time series. *piControl* and *antwater 60S* are shown in red and green respectively. Solid lines show results smoothed with a 10-yr running average.

The relative strength of the AMOC, AABW and AMOC have an impact on zonal mean heat and salinity transport. Overall, meltwater driven changes to ocean circulation act to decrease northward salinity transport between 950-1150, a period of time

in which the AMOC experiences both weakening and strengthening (Fig. 11b). The largest change in salinity transport is at -60°S with a decrease of 0.65 kg/s (Fig. 11b). This corresponds with significant weakening of the AABW observed within these 200 years. However, an overall increase in northward heat transport is observed between 950-1150 (Fig. 11d). This increase in northward heat transport is simulated across most latitudes, with the exception of between -60°S and -50°S (Fig. 11d). It is important to note that during periods of intense AMOC weakening (e.g. 950-1000), a decrease in northward heat transport in the northern hemisphere is observed (Fig. S6). In contrast, when the AMOC is momentarily stronger (e.g. 1100-1150), an increase in northward heat transport is observed (Fig. S7). Changes in both zonal mean salinity and heat transport are driven primarily by the overturning component, with the diffusive and gyre components playing a less significant role (Fig. 11). This is most notable between -90°S and -60°S, whilst in the Northern Hemisphere the overturning component plays a less significant role in driving changes in zonal mean salinity and heat transport. This is reflective of the SO meridional overturning circulation (MOC) having a greater response to Antarctic meltwater than the AMOC.

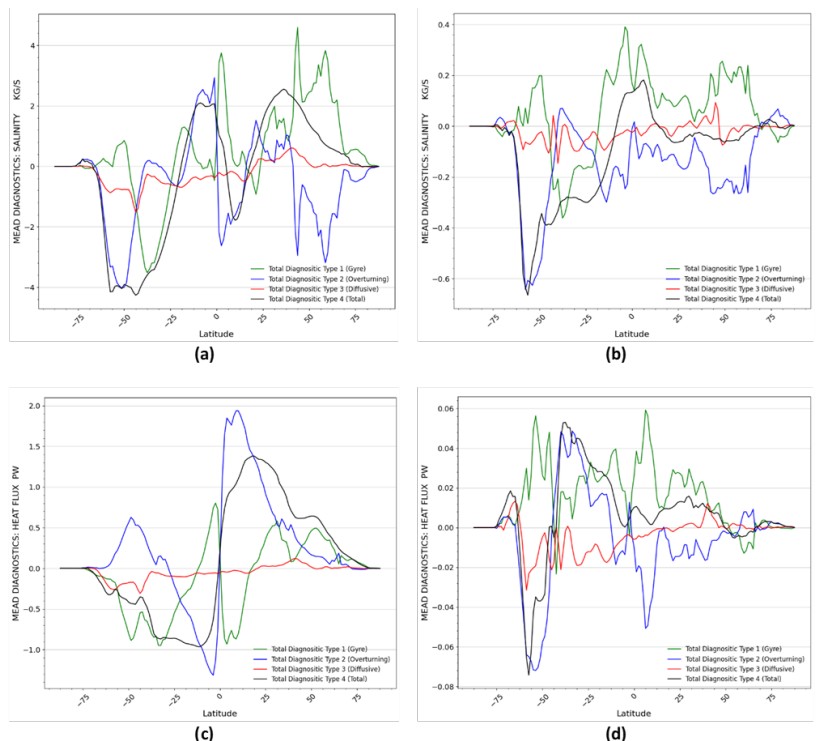

**Figure 11.** (a) Salinity transport (kg/s) partitioned into type for *antwater 60S* in 950-1150. (b) Salinity transport anomaly (kg/s) partitioned into type for *antwater 60S* relative to *piControl*. (c-d) Same as above for northward heat transport (PW).

## 3.3 Hydrological cycle response

Delayed warming in the Southern Hemisphere and increased northward heat transport associated with a slowdown of global overturning circulation results in changes to global precipitation patterns (Sadai et al., 2020; Bronselaer et al., 2018). Notably, a slight northward shift in the ITCZ is simulated, shown as precipitation in the Northern Hemisphere minus the Southern Hemisphere (Bronselaer et al., 2018; Purich and England, 2023) (Fig. 12a). Regional changes in precipitation are also simulated; by the end of the 1150 years, anomalous drying is simulated over much of the SO and the AIS (Fig. 12d). In addition, further anomalous drying is simulated in land regions that are predicted to dry under future climate change, namely eastern Australia. There is some evidence of drying in South America, although this is largely statistically insignificant. There is also evidence for dipole change in precipitation across the Indian and Pacific oceans, with drying simulated in the eastern regions of each ocean (Fig. 12b-d).

However, the difference in annual mean precipitation across the two hemispheres is minimal and highly variable (Fig. 12a). As a result, simulated regional changes are also minimal and do not reflect a drying of the SH and wetting of the NH that is reported by Bronselaer et al. (2018). Our findings are instead more in line with Purich and England's (2023) ANT-MW-LOW experiment, who reported insignificant changes to global precipitation patterns. However, it is important to note that higher meltwater experiments generally report considerably greater changes in the position of the ITCZ, with Bronselaer et al. (2018) and Purich and England (2023) ANT-MW-HIGH experiment reporting a difference in hemispheric precipitation of 0.27 mm/day and 0.22-0.29 mm/day by the end of the century respectively.

## 4 Sensitivity Study

### 4.1 Impact of horizontal distribution

Overall, the results from *antwater* compared with *antwater 60S* suggest that the global climate is not sensitive to the horizontal distribution of freshwater input from the AIS. The global SSS response to AIS meltwater in antwater largely matches that simulated in antwater 60S; annual mean global SSS decreases by  0.15 psu after 1150 years of continuous AIS meltwater (Fig. 13). At depth, results from the antwater experiment vary slightly to *antwater 60S*; the increase in salinity simulated at extreme depths is greater (up to +0.02 psu in the SO) and persists for a longer period of time (Fig. 14). This delay in freshening at depth is likely due to the fact that restricting AIS meltwater to the coastline creates a freshwater cap at the ocean surface, preventing convection and inhibiting deep water formation. This is reflected in a greater reduction in MLD around the AIS coastline in *antwater* compared to *antwater60S* (Fig. 15). This directly inhibits mixing, resulting in more pronounced salinification at depth. This explains the delay in freshening at depth in the global ocean, and why freshening is not observed throughout the entire water column in the SO by the end of the simulation in *antwater* (Fig. 14).

With regards to SATs, the results from *antwater* also largely match those simulated in *antwater 60S* (Fig. 16). However, anomalous warming is simulated in the northern most parts of the Arctic Ocean in *antwater*, and this area of warming extends as far equatorward as 60°N by 1100-1150, with the Greenland Sea experiencing the greatest warming (up to +1.5°C) (Fig.

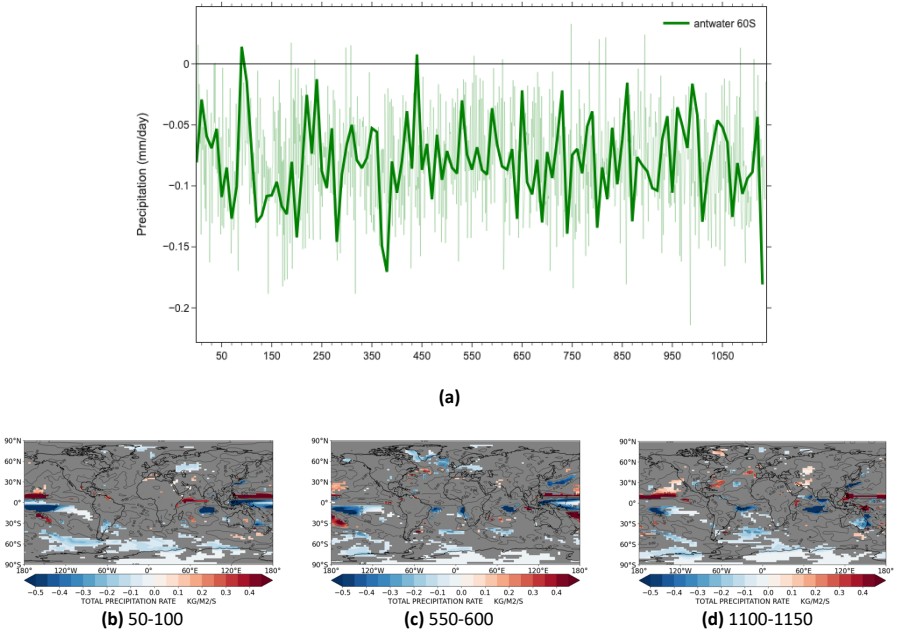

**Figure 12.** (a) Annual mean precipitation in the NH ITCZ band minus the SH ITCZ band time series for *antwater 60S*. Solid lines show results smoothed with a 10-yr running average. (b) 50-100, (c) 550-600 and (d) 1100-1150 meltwater-induced anomalies in precipitation (mm/day) for *antwater 60S* relative to *piControl*. Grey area indicates less than 95% significance

16). In response to this, sea ice thinning is also simulated throughout most of the Arctic Ocean, with thinning of up to 0.5m simulated in the Greenland Sea (Fig. 17).

At depths below 2500m, a change in the horizontal distribution of AIS meltwater results in a greater increase in annual mean global and SO subsurface temperatures (Fig. 18). However, this difference is negligible; at the end of the 1150 years, annual mean global temperatures at depths of 2500m are only 0.05°C warmer in *antwater* compared to *antwater 60S* respectively (Fig. 18a). In addition, confining AIS meltwater input to the coast does not have an impact on either the AMOC or AABW (Fig. 19).

Previous studies have found that the inclusion of iceberg forcings can influence the global climate response to AIS meltwater. Increased oceanic heat loss due to iceberg melting can lead to an increase in the surface cooling effect, and reduce the subsurface temperature feedback amplitude (Schloesser et al., 2019). Merino et al. (2016) found that Antarctic iceberg melt over the SO results in an increase in sea ice cover and a decrease in sea surface temperatures over most of the SO. However, our findings support the work carried out by Pauling et al. (2016), who also concluded that the overall response of the ocean and sea ice is insensitive to a change in horizontal distribution. It is important to note that, to date, there have been few studies that investigate the effect of a change in horizontal distribution of AIS meltwater on the climate response. As such, there remains much uncertainty surrounding this issue.

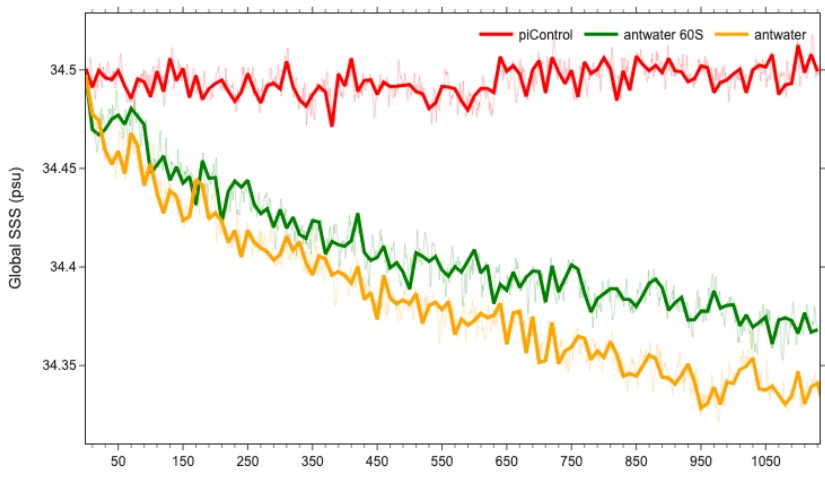

**Figure 13.** Annual mean global SSS time series. Annual mean *piControl*, *antwater 60S* and *antwater* are shown in red, green and orange respectively. Solid lines show results smoothed with a 10-yr running average.

## 5    Discussion and Conclusions

The results presented here show that AIS meltwater causes; freshening of the global ocean at the surface and at depths greater than 5000m; an abrupt reduction in global SATs by up to 1°C that is observed at the moment of freshwater input; a large increase in Antarctic sea ice thickness and growth; an increase in global subsurface temperatures by more than 1°C at depths of 500m-5000m; and a slight northward shift of the ITCZ. Additionally, AIS meltwater also causes a weakening of global overturning circulation, with the AMOC and AABW both experiencing a sustained reduction in strength and the ACC experiencing initial

weakening followed by recovery.

This global climate system response to AIS meltwater is largely due to the fact that AIS meltwater has a significant impact on global overturning circulation. Stratification of waters surrounding the AIS caused by the continuous injection of fresh water forces a slowdown of the AABW and AMOC, as well as a reduction in the upwelling of warmer subsurface waters to the surface (Purich and England, 2023). This causes heat to remain at depth, resulting in the warming of subsurface waters and

cooling at the sea surface which causes cooling of air surface temperatures. An expansion and thickening of sea ice is likely both a response to and cause of cooler surface temperatures and enhanced surface freshening, with the ice-albedo feedback further contributing to cooler SATs (Noble et al., 2020). A weakening of the AMOC induced by AIS meltwater also acts to increase northward heat transport in the Atlantic Ocean, which results in a hemispheric temperature gradient increase that drives a slight northward shift in the (ITCZ) (Sadai et al., 2020; Bronselaer et al., 2018). Since both the AMOC and the AABW

are affected, these changes are observed in both hemispheres. However, the greatest magnitude of change is generally observed in the SO, as AIS meltwater has a greater impact on the AABW than the AMOC.

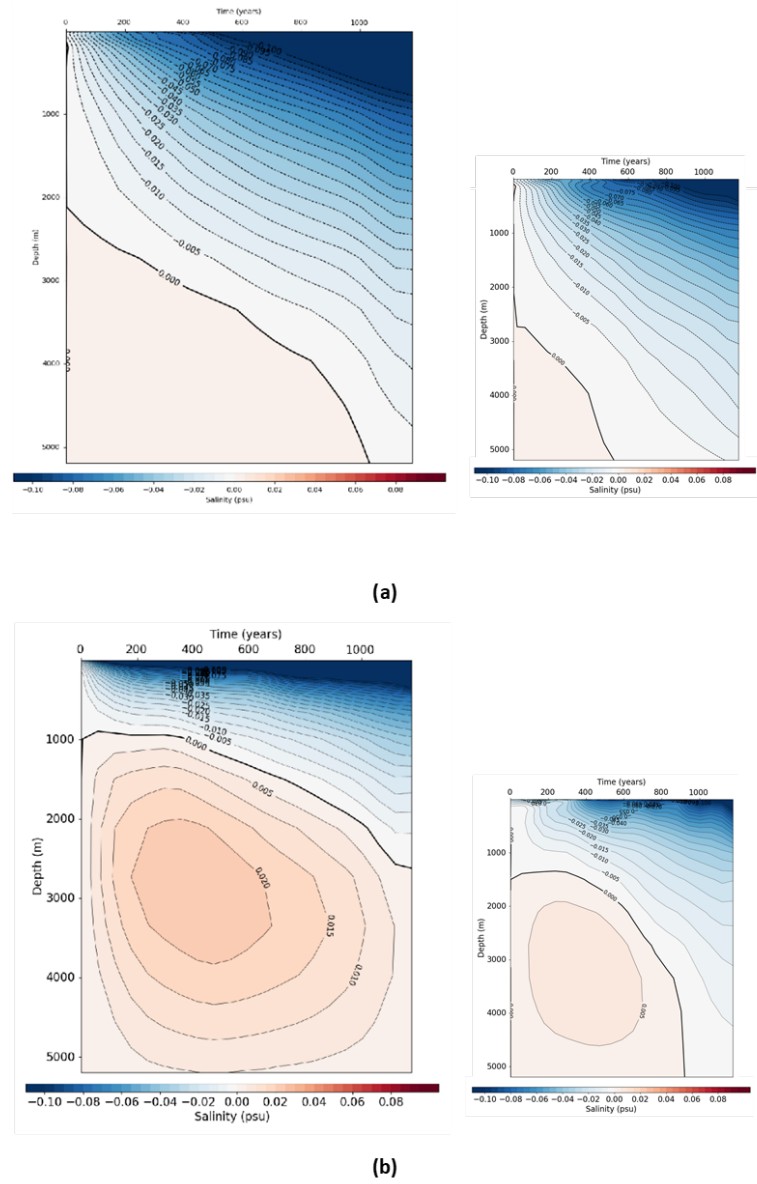

**Figure 14.** (a) Hovmuller diagram (depth (m) vs time) of the anomalous global ocean salinity (psu) for *antwater* relative to *piControl*. *antwater 60S* relative to *piControl* is shown on the right for comparison. (b) Hovmuller diagram of the anomalous SO salinity (psu) for *antwater* relative to *piControl*. *antwater 60S* relative to *piControl* is shown on the right for comparison. Solid black line represents the point at which the salinity anomaly changes from positive to negative.

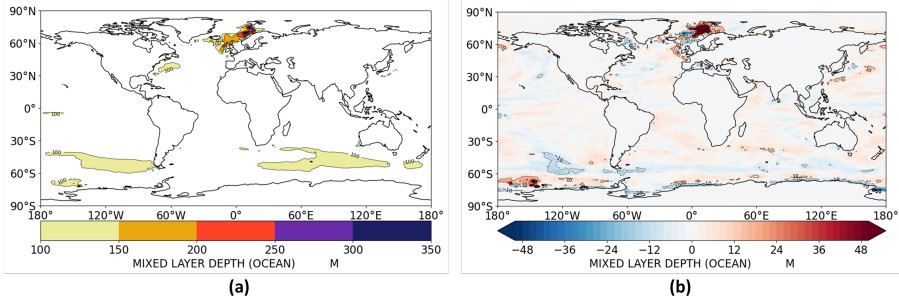

**Figure 15.** (a) Map of annual mean mixed layer depth (MLD) from *antwater* and (b) global anomalies in MLD for *antwater* relative to *antwater 60S* between 1100-1150. Deep water formation simulated in the northern hemisphere occurs during austral summertime, and deep water formation simulated in the southern hemisphere occurs during austral wintertime.

The results produced here provide insights into the global climate implications of rapid AIS melt expected within this century and beyond. In particular, when observing the global climate response to AIS meltwater over relatively long timescales, the global climate system tends to exhibit an initial rapid response before stabilising at a new equilibrium in which ocean currents

are weaker, SATs are cooler and subsurface temperatures are warmer. Additionally, the global climate system is relatively insensitive to a change in horizontal distribution. Concentrating freshwater input around the Antarctic coastline (as in *antwater*) results in the same climate response as releasing freshwater across the entire SO (as in *antwater 60S*). This supports the notion that the manner in which the AIS loses mass, whether that be predominately through iceberg mass loss or basal melt, is unlikely to affect the climate response.

This project is not without limitations. As mentioned previously, ice sheets are not modelled dynamically in HadCM3-M2.1. As such, we are unable to assess the impacts of AIS meltwater on the ice sheet itself. In addition, like many CMIP6-class coupled climate models, HadCM3-M2.1 is unable to directly resolve several complex processes including mesoscale and submesoscale dynamics, the continental slope current, and deep sea water (DSW) overflows (Valdes et al., 2017; Swart et al., 2023). Although higher-resolution and more complex climate models are therefore likely to be better at resolving these

dynamics, HadCM3-M2.1, as an efficient fully coupled AOGCM, is still an appropriate model to help better understand the long term climate response to AIS meltwater. Additionally, in all experiments the additional freshwater flux is applied at the ocean surface. It is important to note that this is not representative of how meltwater from the AIS would enter the ocean, as meltwater in the form of basal melt would enter the ocean at depths of several hundred metres (Bronselaer et al., 2018). In addition, releasing meltwater at depth can affect the ocean mixed layer; Pauling et al. (2016) found that injecting meltwater

at the depth of ice shelf bases around Antarctica caused the ocean mixed layer to deepen, while adding freshwater at the ocean surface caused the mixed layer to shoal (Pauling et al., 2016). However, the depth at which meltwater is released in climate models does not appear to affect the modelled sea ice response (Pauling et al., 2016; Bronselaer et al., 2018; Golledge et al., 2019). As such, whilst we acknowledge that applying a freshwater flux at surface level is a limitation, it is unlikely to have a significant impact on the modelled climate response (Bronselaer et al., 2018). In this study, we investigate the modelled

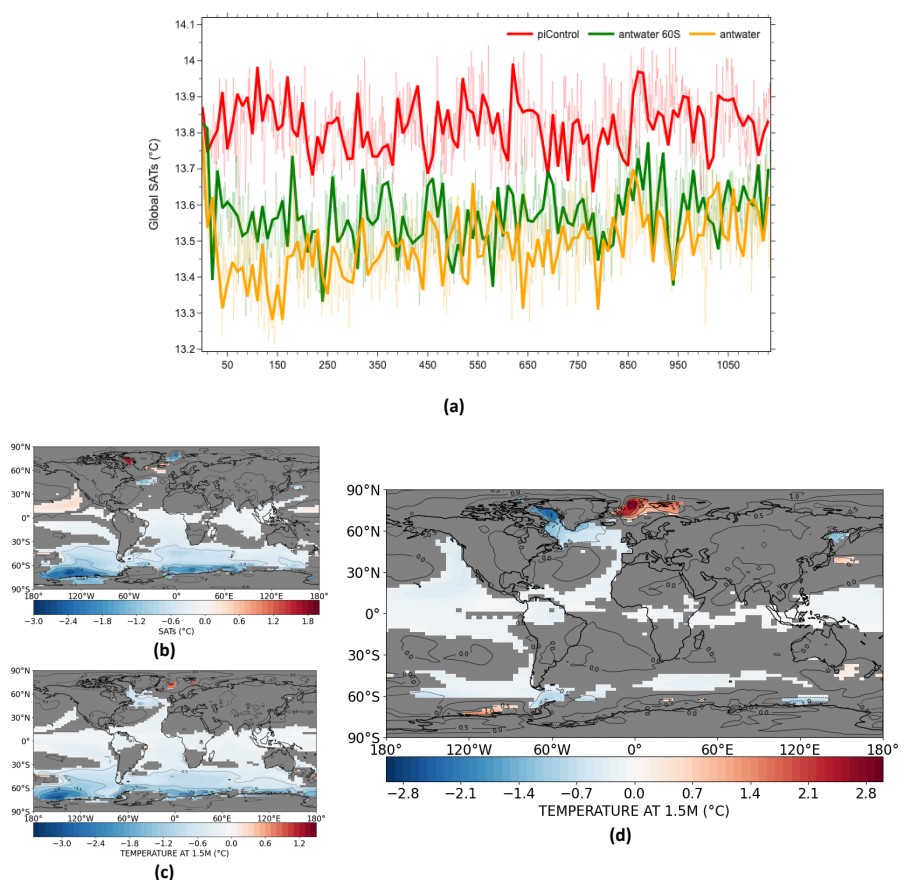

**Figure 16.** (a) Annual mean global SAT time series. Annual mean *piControl*, *antwater 60S* and *antwater* are shown in red, green and orange respectively. Solid lines show results smoothed with a 10-yr running average. 1100-1150 anomalies in SATs (°C) for; (b) *antwater 60S* relative to *piControl*, (c) *antwater* relative to *piControl* and (d) *antwater* relative to *antwater 60S*. Grey area indicates less than 95% significance

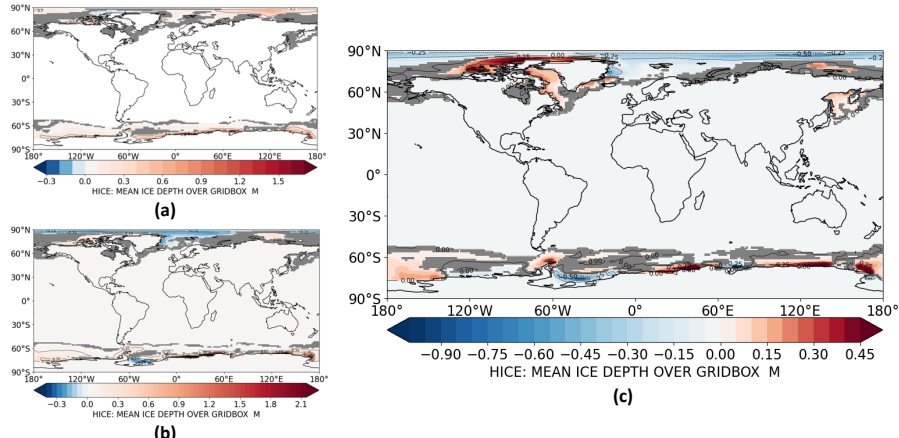

**Figure 17.** 1100-1150 anomalies in global sea ice thickness (m) for; (a) *antwater 60S* relative to *piControl*, (b) *antwater* relative to *piControl* and (c) *antwater* relative to *antwater 60S*. Grey area indicates less than 95% significance

climate response to AIS meltwater exclusively, and therefore do not include freshwater perturbations from other major sources, including the GIS. As discussed previously, the AMOC is more sensitive to changes in GIS meltwater than AIS meltwater, and as such, incorporating meltwater from the GIS into our experiments would likely have a substantial impact on the global climate response.

Future work concerning the climate response to AIS collapse could revolve around taking a more 'realistic' rather than idealised approach, akin to the Tier 2 experiments proposed by the SOFIA Initiative (Table A1) (Swart et al., 2023). This would involve incorporating GIS meltwater, or altering the horizontal distribution of freshwater to a state that is most likely to occur in the near future, such as adding freshwater exclusively to the Amundsen and Bellingshausen Seas, an area projected to experience abnormally high melt rates compared to the rest of the AIS. Additionally, as discussed previously, altering the vertical distribution of freshwater flux would also create a more realistic representation of how AIS meltwater would enter the SO. This approach would also involve applying a more plausible freshwater flux based on historical and future rates of freshwater input, as well as including additional climate forcings, such as increasing greenhouse gas concentrations or incorporating climate forcing scenarios (e.g. SSPs). This approach would allow us to develop our understanding of how the inclusion of AIS meltwater influences historical simulations and future projections (Swart et al., 2023). Additional experiments could also involve altering the freshwater flux under a given emission scenario to assess the climate response to different magnitudes of AIS collapse. Previous studies have shown that the global climate system responds nonlinearly to meltwater addition (Park and Latif, 2019; Schloesser et al., 2019; Purich and England, 2023). However, this remains a clear gap in our understanding. As such, considering the large uncertainties surrounding the processes involved in AIS mass loss and the overall stability of the AIS, investigating the impact of altering the freshwater flux on the global climate system would allow us to best prepare for all potential future scenarios.

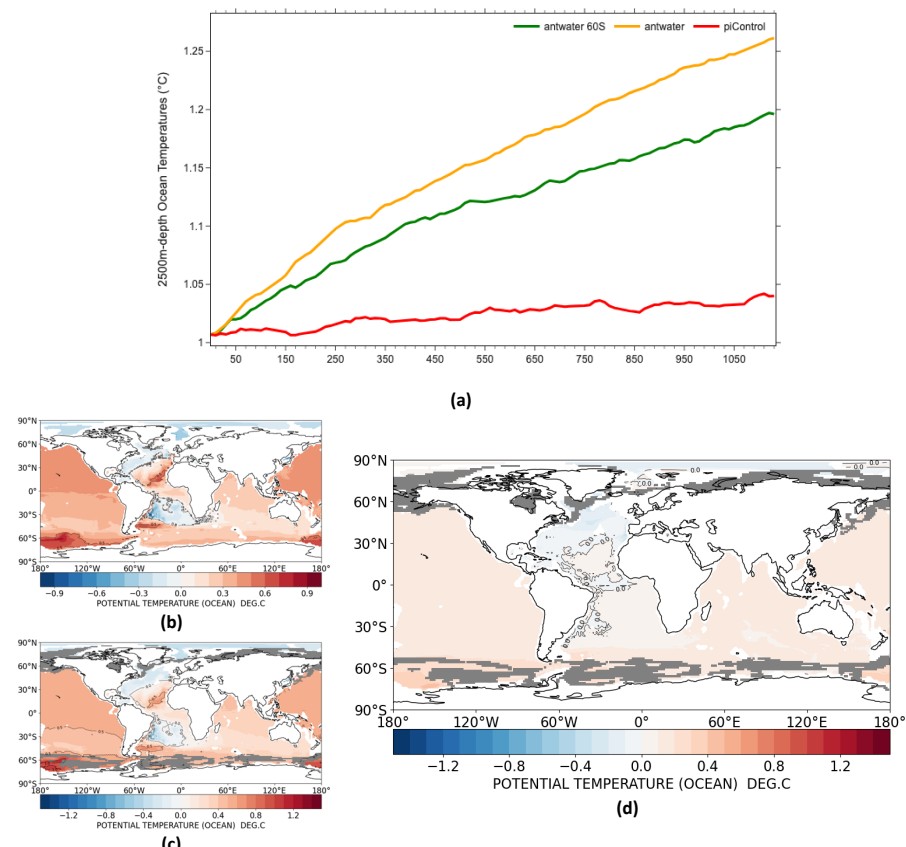

**Figure 18.** (a) Annual mean global subsurface temperatures (2500m) (°C) time series. Solid lines show results smoothed with a 10-yr running average. 1100-1150 anomalies in global subsurface temperatures (2500m) (°C) for; (b) *antwater 60S* relative to *piControl*, (c) *antwater* relative to *piControl* and (d) *antwater* relative to *antwater 60S*. Grey area indicates less than 95% significance

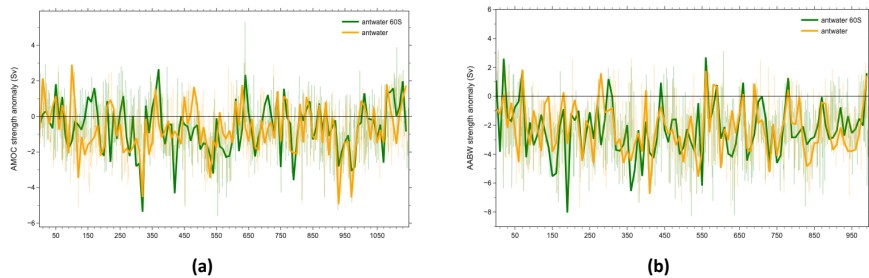

**Figure 19.** (a) Annual mean AMOC strength (Sv) anomaly and (b) Annual mean AABW strength (Sv) anomaly time series for *antwater* relative to *piControl* (orange) and *antwater 60S* relative to *piControl* (green). Dashed lines show results smoothed with a 10-yr running average.

*Data availability.* Archive of key variables can be found here; https://www.paleo.bristol.ac.uk/ummodel/scripts/papers/

*Author contributions.* AM wrote the first draft, designed the experiments and carried them out. DL contributed to the discussion of the experimental design and the editing of the text. XR assisted with developing the code required and running the initial simulations.

*Competing interests.* The authors declare that they have no conflict of interest.

*Acknowledgements.* This work was carried out in the framework of the University of Bristol Cabot Institute for the Environment MScR in Global Environmental Challenges.

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
