# Peer review of "Fig. S1. Annual mean global salinity weighted with depth (psu) time series for piControl (red) and antwater 60S (green). Dashed grey line shows expected changes in global salinity weighted with depth based on projected input of freshwater."

_EGUsphere, 2025_

## Author Comment (AC1)

First of all, many thanks for your detailed comments, which will enable us to significantly improve our manuscript.

Our point-by-point responses are detailed below.

The original review comments are shown in red and our responses in **black**.

1. Title: Perhaps including "in HadCM3-M2.1" would help to clarify the scope for readers?

**A: We agree with the suggestion and will add "in HadCM3-M2.1" to the revise paper title.**

2. 3: >"the effects of AIS meltwater are not considered by most existing coupled climate models": you do have to be careful here, because ice sheet dynamics are not considered by these models, but most if not all of them do permit surface melt which does add additional freshwater under warming (indeed, this can be an infinite source of freshwater in many models).

**A: We will rewrite this sentence to clarify that ice-climate feedbacks are not considered by most existing coupled climate models (Swart *et al.*, 2023).**

3. 18-25: Between para's 1 and 2 there is an implicit link between sea level rise and meltwater input to the ocean. Just noting that these are not strictly the same thing (because melt of floating iceshelves is a freshwater forcing, that does not influence SLR). Not a huge deal, more of a note.

**A: Following this suggestion, we will rephrase these paragraphs to emphasise the importance of non-iceshelf components in contributing to SLR.**

4. 40: It is also important to use a consistent design across the community - which you are contributing to (and is a valuable part of the paper) - I note this is mentioned near ln 70

**A: We agree that using a consistent experimental design is an important and valuable aspect of this study. To further emphasise this, we will add the following sentence to paragraph ln65-70;**

**"As mentioned previously, inconsistent experimental design across the existing literature is creating uncertainties over the magnitude of the global climate system's response to AIS meltwater, inhibiting our ability to constrain the climate impacts of AIS meltwater. By drawing upon and contributing to the SOFIA Initiative, this study aims to improve our understanding of the global climate response to AIS meltwater through the use of a common experimental design."**

5. 54: I wonder if there is a slightly more constructive way to frame this? Such as "it's unclear whether results from simplified EMICs would hold in fully coupled AOGCMs".

**A: We agree with this suggestion, and will rewrite line 54 to; "However, it is unclear whether results from simplified EMICs and ocean-only models would hold in fully coupled AOGCMs, as these models do not fully account for important atmospheric or cryospheric feedbacks …"**

6. 108-109 / 119: It would be useful to provide more detail on the converting of the FW flux to a virtual salt flux, since this is presumably the same mechanisms used to implement the SOFIA experiments (not mentioned on 119 but should be).

**A: a) We will provide more detail on how freshwater fluxes are represented as virtual salt fluxes in the HadCM3-M2.1 model. The following will be included in section 2.1 Model Description;**

**"As HadCM3-M2.1 uses a rigid-lid, the ocean surface is not allowed to deform and, as a result, total ocean volume remains constant. As such, any freshwater fluxes simulated in the model do not physically increase total ocean volume, but are instead applied as virtual salinity fluxes that reduce ocean surface salinity. These virtual salinity fluxes effectively remove salt mass from the model ocean, mimicking the dilution effect that would occur with a physical ocean volume increase."**

**b) Following this, we will clarify that the freshwater fluxes added in "antwater" and "antwater60S" are implemented in same way as detailed above, and conform with experimental designs detailed by the SOFIA initiative.**

> 7. 121: >"although run over a longer time scale (Table S1)" - the referenced experimental design specifies a length of >=100 years. It is good that this work exceeds the minimum, but its seems entirely consistent with the proposed design.

**A: We will rewrite ln 121 to clarify that our experiments are consistent with the SOFIA initiative's "antwater60S" and "antwater" experiment designs.**

> 8. 125: Please state what the magnitude of this climatological flux is. In most similar models, this flux is roughly specified to balance P-E over Antarctica. If that is true, it is not a small flux.

**A: We will include the magnitude of the base climatological flux included in all experiments.**

> 9. 143: As written, this does not seem to quite make sense to me. A positive salinity trend cannot be due to (positive down) freshwater flux. A positive salinity trend might indicate that the piControl freshwater flux is smaller than P-E over Antarctica, and hence, P-E-R flowing over the ocean is less than 0 (i.e. effectively a negative freshwater flux). That is assuming everything else in the model is conserving of freshwater, which is not clear.

**A: We will rewrite ln 141-143 to provide further clarity over the role of snow build-up in creating the positive salinity trend simulated in the "piControl" experiment. We will explain how the background flux applied in the HadCM3-M2.1 model is not quite enough to completely balance the build-up of snow on ice sheets simulated in the model, as the background flux is a fixed field, rather than being prognostically calculated (Valdes *et al.*, 2017). Alongside a more detailed explanation of the virtual salinity flux detailed in Comment #6, these changes should provide clarity over the positive salinity trend simulated by the model.**

> 10. Fig 3a/b and timeseries plots in general: The annual scale of variability is not really discussed in this manuscript, and it is very noisy. I think the results would be clearer is the annual mean lines were de-emphasized, and the 10 yr running averages were made more prominent.

**A: We agree that annual variability is difficult to read, and so we will de-emphasise the annual mean lines and emphasise the 10-year running averages in all timeseries plots.**

> 11. 170 and surrounding. >"in line with observations from the same time period shown here". I find using the term "observations" confusing (with actual real world data). Also, the magnitude of freshening is compared with other studies, but it is not mentioned how the rates of forcing match or do not. If the freshening is similar, but the forcing is very different, this does not indicate a consistency. Please clarify.

**A: a) Following your suggestions, we will replace the term "observations" with "simulations". b) We will state that the experiment run by Li *et al.*, (2023) added 0.08 Sv of AIS meltwater for 50 years under a RCP8.5 scenario. As such, we will clarify that whilst our experiments use a similar freshwater forcings, the Li *et al.*, (2023) study is likely influenced by additional forcings.**

> 12. 179: Is that the Labrador Sea? The map is small, but it looks more like Baffin Bay or Davis Straight to me.

**A: We agree with your suggestion, and will correct ln 179 to say "Baffin Bay".**

> 13. 193: Global sea-ice thickness is not really such a useful metric, given the massive climatogical thickness differences across the hemispheres. Perhaps reporting as a change per NH/SH, and perhaps even a relative (%) change would be clearer?

**A: We agree with this suggestion, and we will report changes in sea ice thickness in the NH and SH in addition to global sea-ice thickness.**

> 14. Figure 6 c-h: I think it would be more informative to saturate the high end of the colorbar, so that the broader pattern of thickness increase is visible. As it stands, we only see what happens near the edges. This is subjective, but I feel like we are missing the broader picture.

**A: Following your suggestion, we will adjust the colour bars of Figures 6 c-h.**

15. 235: What is the climatological AMOC rate in this model (in piControl)?

**A: We will report the climatological AMOC rate in "piControl" in the revised paper.**

16. 230-260: It might be worth testing whether the AMOC/ACC timeseries are statistically different. As you say, there is a lot of variability.

**A: We agree with this suggestion, and will conduct two-way student t-tests on the AMOC, AABW and ACC timeseries data to determine statistical significance at the level 95%.**

17. Figure 12: The changes in mid latitude precip are lost in the large changes near the equator. If you present the precip change as relative to the piControl (% as opposed to mm/day), it will take care of the climatological differences across latitudes, and perhaps be more meaningful (again, subjective).

**A: Following your suggestion, we agree that mid latitude precipitation is lost in the large changes near the equator. To resolve this, we will adjust the colour bars of Figure 12 b-d to more clearly show climatological differences across latitudes.**

18. 290: this explanation does not make physical sense to me, or I'm missing something. If injecting water around the coast makes it "able to spread horizontally across the surface of the SO before diffusing into the ocean depths", then injecting it even south of 60S would surely enable even more spread. Might a possible explanation might be that injection directly around Antarctica in antwater influences SO deep convection more directly than antwater60S, leading to more pronounced salinification at depth. I know Fig 18 shows similar AABW responses, but deep convection changes can be quite different to AABW (e.g. Chen *et al.*, 2023)

**A: We agree with the explanation that you have provided here, and would like to thank you for this helpful contribution to our paper. Having revisited our work, we now argue that restricting AIS meltwater to the coastline creates a freshwater cap at the ocean surface, preventing convection and inhibiting deep water formation. This directly inhibits mixing, resulting in more pronounced salinification at depth. To confirm this, we will edit Figure 9 to also show a map of annual mean mixed layer depth (MLD) from "antwater", to assess whether a change in horizontal distribution results in a change in the rate of downwelling around the AIS coastline.**

19. 295: Could this just variability, even in a 50 year mean?

**A: We argue that the results from the two-way student t-test indicate that the anomalous warming simulated in the northern parts of the Arctic Ocean is statistically significant, and therefore unlikely to be a result of variability.**

20. 314: This "significant impact" statement seems to contradict the earlier results that AMOC changes are very small, and even AABW changes are not huge (but we do not know the relative changes). I would contend that the stratification mechanism that you mention next is likely more important.

**A: We will first conduct statistical significance tests on the AMOC and AABW timeseries, and will amend ln 314 depending on our findings. If our results indicate that AMOC and AABW changes are in fact statistically significant, this will be reflected in our conclusions.**

21. 319: Sea-ice changes vs SAT changes are a bit chicken and egg. How much surface cooling could be explained by the ice-albedo feedback alone?

**A: We agree with this comment, and we will rewrite ln 319 to reflect the dynamic relationship between sea ice change and SAT changes; "An expansion and thickening of sea ice is likely both a response to and cause of cooler surface temperatures and enhanced surface freshening, with the ice-albedo feedback further contributing to cooler SATs".**

22. 359: DATA availability: The authors are describing the results of running a standardizes experimental protocol in their model. The SOFIA project has a data archive and active working group, but it would seem this data has not been contributed to that open data archive. It would be added value to contribute that data:https://sofiamip.github.io/data-access.html

**A: We are happy to share our data with the SOFIA project, and will contribute to SOFIA's open data archive.**

*References:*

Chen, J.-J. *et al.* (2023) "Reduced Deep Convection and Bottom Water Formation Due To Antarctic Meltwater in a Multi-Model Ensemble," *Geophysical Research Letters*, 50(24), p. e2023GL106492. Available at: https://doi.org/10.1029/2023GL106492.

Li, Q. *et al.* (2023) "Abyssal ocean overturning slowdown and warming driven by Antarctic meltwater," *Nature*, 615(7954), pp. 841–847. Available at: https://doi.org/10.1038/s41586-023-05762-w.

Swart, N. *et al.* (2023) "The Southern Ocean Freshwater release model experiments Initiative (SOFIA): Scientific objectives and experimental design," *EGUsphere*, 2023, pp. 1–30. Available at: https://doi.org/10.5194/egusphere-2023-198.

Valdes, P.J. *et al.* (2017) "The BRIDGE HadCM3 family of climate models: HadCM3@Bristol v1.0," *Geoscientific Model Development*, 10(10), pp. 3715–3743. Available at: https://doi.org/10.5194/gmd-10-3715-2017.

---

## Author Response (AR1)

We thank the two reviewers for their very helpful comments on the manuscript, which have greatly improved the paper.

Here we provide a point-by-point response to those comments. The line numbers **in blue** refer to the marked-up ('latexdiff') version of the manuscript, which indicates how this revised manuscript compares to the originally submitted version.

**Reviewer #1**

line 27: you mention that previous studies used 'idealised freshwater hosing experiments', but don't clarify what you mean by 'idealised'. This is important, because your study uses a low-resolution highly-parameterised GCM and imposes a constant salinity flux as a proxy for meltwater. This is what I would call 'idealised', since the meltwater flux is entirely fictional, it does not derive from any ice sheet model. By contrast, the papers you cite in lines 30-31 do a far more robust job with the meltwater - Sadai et al used a coupled ice sheet - climate model, Golledge et al used an offline-coupled ice sheet - climate model, and Bronselaer used meltwater fluxes from a previous ice sheet simulation (DeConto & Pollard, 2016). In all three cases, the meltwater added to the climate model comes from a dynamically-evolving ice sheet model, and, in the studies of Sadai and Golledge, that ice sheet evolution is in turn coupled to the evolution of the simulated climate. I wouldn't therefore call them 'idealised'.

**We have amended this line in the revised paper to 'freshwater perturbation experiments'. In addition, we have included a definition of an idealised freshwater hosing experiment in the "Experiment Design" section of the paper to highlight the differences in experimental design between ours and others who have carried out model-informed meltwater experiments.**
**See Section 1; line 28 and Section 2.2; lines 133-135.**

line 32: again, you mention 'interactive ice sheets and shelves' as being absent from previous work, but they are also absent from your model, so...?

**We have expanded on this point in the revised paper by explaining that the exclusion of interactive ice sheets and shelves represents a lack of inclusion of the impacts of ice-climate feedbacks from coupled climate model simulations included in CMIP6. We have also explained that this paper aims to address this gap in the literature by conducting a series of idealised freshwater hosing experiments that incorporate the impact of these feedbacks (without actually representing the processes themselves).**
**See Section 1; line 33 and Section 1; line 69.**

line 46-7: I assume here you mean future projections? Because there are several studies that consider multi- centennial to multi-millennial hosing under palaeo scenarios (Weaver et al., 2003; Menviel et al., 2010; Bakker et al., 2017; for example).

**We have modified this line to clarify that, to date, there have been few studies that project future climate responses to freshwater flux from the AIS beyond a few hundred years.**
**See Section 1; lines 47-48.**

line 63-4: Noble et al and Bentley et al are both review papers, and PD09 do not mention meltwater pulses, only ice growth and decay, so why were these references chosen specifically? There are ice sheet simulations published that explicitly focus on AIS meltwater pulses (Stuhne & Peltier 2015; Golledge et al., 2014), so perhaps the referencing could be broadened to at least acknowledge where original work on this topic has been done?

**Following your suggestion, we have explored and included studies that explicitly focus on AIS meltwater pulses, including Golledge et al. (2014) and Stuhne and Peltier (2015).**
**See Section 1; lines 67.**

line 68-70: "The results produced here will provide novel insight into the global climate implications of rapid AIS melt expected within this century and beyond" - I'm not sure how well the study succeeds in providing novel insights - perhaps a summary could be added here or later that identifies what the study shows that is consistent with existing knowledge, and what has been found that is new?

**Following your suggestion, we have provided a summary of what insights this paper offers into the global climate implications of rapid AIS melt in the Conclusion section of the manuscript.**

**See Section 5; lines 377-384.**

line 95-102: there is a long list here of the various parameterisations employed, but I didn't find a discussion of how these approximations might influence the results? Are there studies that could be referenced here that have shown eg that eddy-resolving models show xx greater or lesser mixing / transport / whatever? I feel like some justification / explanation is needed here.

**We have included a new paragraph describing the use of a tuned version of the HadCM3-M2.1 model, as described in Ren et al. (2023). We have also included a new table in the supplementary material (Table S3) that details the impacts of these tuned changes on the model results. We have also expanded upon the model description to include a discussion of how the parameterisations employed may have influenced our results based on the work of Frigola et al. (2025) (in particular, the effects of eddy parameterisation).**

**See Section 2.1; lines 111-114 and Section 2.1; lines 126-130.**

line 105: "excess salt formed by melting" - I'm sorry if I misunderstand what is meant here, but I thought it was the formation of sea ice (ie by freezing) that rejected salt?

**We agree with the suggested changes, and have modified this line to; "Sea ice is assumed to have a constant salinity, with salt/freshwater fluxes formed by sea ice formation/melting represented as virtual salinity fluxes into the ocean".**

**See Section 2.1; line 117-119.**

line 111 and throughout: why the use of future tense here? The work has been done, so either use past tense or present.

**We have corrected this section to use the present tense instead of the future tense to maintain consistency throughout the paper.**

**See Section 2.2; lines 144-155.**

line 119: Regarding the addition of meltwater at the surface, this is fine and a common approach, but maybe mention the limitation of it. For example, Pauling et al 2016 showed that releasing meltwater at the depth of ice shelf bases can have an influence on its impact: "We found that injecting water at the depth of the front of ice shelf around Antarctica caused the ocean mixed layer to deepen, while adding freshwater at the surface caused the mixed layer to shoal."

**Following your suggestion, we have expanded the "Discussion and Conclusions" section to include more details about the limitations of adding meltwater at the surface level instead of at depth.**

**See Section 5; lines 393-396.**

line 126: 'perturb from' sounds odd, maybe 'branch from'?

**Changed.**

**See Section 2.2; line 139.**

line 128: "The difference between the piControl and FW experiments taken as modelling responses." - seems like some words are missing from this sentence.

**Fixed.**

**See Section 2.2; lines 141-142.**

line 132: "at the level 95%" --> "at the 95% confidence level" perhaps?

**Changed.**

**See Section 2.2; line 165.**

line 139-140: "To date, this is the first study to investigate the climate response to AIS meltwater using a fully coupled climate model over such long time scales." - Given the limitations of the model and methodology, I would suggest tempering this statement a little. Other studies as identified above have done similar things, for example Bakker et al used actual ice sheet meltwater fluxes and ran for 5000 years.

**We have removed this statement from the manuscript.**
See Section 2.2; lines 172-173.

line 148-150: "The area immediately surrounding the AIS experiences a decrease in SSS of up to 0.5 psu (Fig. 3h). Freshening around the AIS is relatively uniform, with no particular region experiencing significantly more freshening (Fig. 3f-h)." - This is the area you apply the meltwater, right? And that is applied as a salinity anomaly? So these aren't really 'results', this is just what you've put into the model.

**We have amended this section by clarifying that freshening simulated around the coastline of the AIS is a result of AIS meltwater being applied directly into this region. However, the amount of freshening in psu is a "result", because it results from a combination of the input salinity flux and the advection of the signal of this flux.**
See Section 3.1; line 186.

line 153: 'weaker rate' - 'lower rate' maybe?

**Changed.**
See Section 3.1; line 189.

line 154: 'This freshening' - always best to avoid using 'this' unless you clarify what it refers to, eg 'The surface freshening of xx psu described above...'

**Fixed.**
See Section 3.1; line 191.

line 166: 'ran' --> 'run'

**Fixed.**
See Section 3.1; line 203.

line 184-5: NB - Golledge et al 2019 also used freshwater fluxes from the Greenland Ice Sheet, which most likely affects N.Atl SSTs etc.

**We have modified this sentence to clarify that the relatively low magnitude of cooling simulated here is to be expected as the magnitude of meltwater input in this study is relatively small, no additional forcings are included and this study only simulates AIS meltwater.**
See Section 3.1; lines 223-224.

line 189-194: You refer to 'this unusual trend' and 'this trend' and an increase in global sea ice thickness of c. 0.1m. But I don't see any trend in any of the data in 6a or 6b. There is a lot of variability, but how have you calculated a thickening trend? And how can the global increase be 0.1m when the plot in 6a shows values of only 0.07 to 0.1? I think the step change between picontrol and the forced run is 0.01m? It is 0.1m in the SO experiment.

**To highlight the trend of the global sea ice thickness data, we have edited Figures 6a and 6b to further emphasise the 10-year running average and de-emphasise the annual mean. In addition, we have clarified how we have calculated a thickening trend in the data by reporting the increase in the global sea ice thickness average between 0-100 years. This demonstrates that the antwater60S simulation rapidly reaches a new equilibrium state. We have also corrected the manuscript by stating that the step change between "piControl" and "antwater60S" is 0.01m as opposed to 0.1m.**
See Section 3.1; line 233, Section 3.1; lines 234-236, and Figure 6a, b.

line 200-1: Is this because the location of meltwater input in Sadai et al is controlled by locations of mass loss in the ice sheet model? Or is there some other reason?

**We have modified this line to; "Sadai et al. (2020) also reported sea ice thickening across the SO, with thickening concentrated in regions in which meltwater is directly added, namely the Amundsen and Bellinghausen Sea and Ross Sea.".**
See Section 3.1; lines 240-241.

line 233: Maybe also look at Golledge et al 2019 for how fluxes from AIS and GIS affect AMOC differently.

**In the Results section, we have included a brief explanation of how fluxes from the AIS and GIS affect AMOC differently. We have also addressed the limitations of excluding GIS meltwater in the Conclusion section.**

See Section 3.2; lines 278-281 and Section 5; lines 399-403.

line 234 and line 236: I don't think you can make the claim at 236 when you just made the statement at 234!

**We have modified this section to ensure consistency.**

See Section 3.2; lines 276-277.

line 285-6: Can you say anything about the effect of icebergs, eg from simulations that explicitly simulated them (eg Schloesser et al)?

**We have added a section exploring the effect of icebergs from studies that explicitly simulate them, including Schloesser et al. (2019) and Merino et al. (2016).**

See Section 4.1; lines 351-354.

line 294-8: Note that this may be different if GIS meltwater was included in the expts.

**We have added a paragraph in the Discussion and Conclusions section that explores the limitations of excluding GIS meltwater from our experiments.**

See Section 5; lines 399-403.

line 309: rather than 'dramatic' I would just say 'abrupt' - this rapid change just reflects your methodology of meltwater addition, correct?

**Changed.**

See Section 5; line 361.

**Reviewer #2**

Title: Perhaps including "in HadCM3-M2.1" would help to clarify the scope for readers?

**Fixed.**

See Title.

3: >"the effects of AIS meltwater are not considered by most existing coupled climate models": you do have to be careful here, because ice sheet dynamics are not considered by these models, but most if not all of them do permit surface melt which does add additional freshwater under warming (indeed, this can be an infinite source of freshwater in many models).

**We have modified this sentence to clarify that ice-climate feedbacks are not considered by most existing coupled climate models.**

See Abstract; line 3.

18-25: Between para's 1 and 2 there is an implicit link between sea level rise and meltwater input to the ocean. Just noting that these are not strictly the same thing (because melt of floating iceshelves is a freshwater forcing, that does not influence SLR). Not a huge deal, more of a note.

**We have rephrased these paragraphs to emphasise the importance of non-iceshelf components in contributing to SLR.**

See Section 1; line 21.

40: It is also important to use a consistent design across the community - which you are contributing to (and is a valuable part of the paper) - I note this is mentioned near ln 70

**We agreed that using a consistent experimental design is an important and valuable aspect of this study, and we have added a paragraph explaining this.**

See Section 1; lines 73-77.

54: I wonder if there is a slightly more constructive way to frame this? Such as "it's unclear whether results from simplified EMICs would hold in fully coupled AOGCMs".

**We have modified this line.**
See Section 1; lines 55-57.

108-109 / 119: It would be useful to provide more detail on the converting of the FW flux to a virtual salt flux, since this is presumably the same mechanisms used to implement the SOFIA experiments (not mentioned on 119 but should be).

**We have provided a more detailed explanation of how freshwater fluxes are represented as virtual salt fluxes in the HadCM3-M2.1 model. In addition, we have clarified that the freshwater fluxes added in "antwater" and "antwater60S" are represented as virtual salt fluxes and are consistent with experimental designs detailed in the SOFIA initiative.**
See Section 2.1; lines 123-126 and Section 2.2; lines 153-155.

121: >"although run over a longer time scale (Table S1)" - the referenced experimental design specifies a length of >=100 years. It is good that this work exceeds the minimum, but its seems entirely consistent with the proposed design.

**We have modified this line to clarify that our experiments are entirely consistent with the SOFIA initiative's "antwater60S" and "antwater" experiment designs.**
See Section 2.2; lines 153-155.

125: Please state what the magnitude of this climatological flux is. In most similar models, this flux is roughly specified to balance P-E over Antarctica. If that is true, it is not a small flux.

**We have added the magnitude of the base climatological flux included in all experiments.**
See Section 2.2; lines 138-139.

143: As written, this does not seem to quite make sense to me. A positive salinity trend cannot be due to (positive down) freshwater flux. A positive salinity trend might indicate that the piControl freshwater flux is smaller than P- E over Antarctica, and hence, P-E-R flowing over the ocean is less than 0 (i.e. effectively a negative freshwater flux). That is assuming everything else in the model is conserving of freshwater, which is not clear.

**We have explained in greater detail the role of snow build-up in creating the positive salinity trend simulated in the "piControl" experiment. We have explained in the Experiment Design section that the background flux applied in the HadCM3-M2.1 model is not quite enough to completely balance the build-up of snow on ice sheets simulated in the model, as the background flux is a fixed field, rather than being prognostically calculated. We have also included a more detailed explanation of the virtual salinity flux in Section 2.1.**
See Section 2.1; lines 123-126, Section 2.2; lines 137-139 and Section 2.2; lines 176-179.

Fig 3a/b and timeseries plots in general: The annual scale of variability is not really discussed in this manuscript, and it is very noisy. I think the results would be clearer is the annual mean lines were de-emphasised, and the 10 yr running averages were made more prominent.

**We have agreed that annual variability is difficult to read and have de-emphasised the annual mean lines and emphasised the 10-year running averages in all timeseries plots.**
See Figures 3, 5, 6, 8, 10, 12, 13, 16, 18 and 19.

170 and surrounding. >"in line with observations from the same time period shown here". I find using the term "observations" confusing (with actual real world data). Also, the magnitude of freshening is compared with other studies, but it is not mentioned how the rates of forcing match or do not. If the freshening is similar, but the forcing is very different, this does not indicate a consistency. Please clarify.

**Following your suggestions, we have replaced the term "observations" with "simulations". We have also stated that the experiment run by Li et al., (2023) added 0.08 Sv of AIS meltwater for 50 years under a RCP8.5 scenario.**
See Section 3.1; lines207-208 and Section 3.1; line 209.

179: Is that the Labrador Sea? The map is small, but it looks more like Baffin Bay or Davis Straight to me.

**We have corrected this to say "Baffin Bay".**
See Section 3.1, line 217.

193: Global sea-ice thickness is not really such a useful metric, given the massive climatogical thickness differences across the hemispheres. Perhaps reporting as a change per NH/SH, and perhaps even a relative (%) change would be clearer?

**We have reported changes in sea ice thickness in the NH and SH in addition to global sea-ice thickness.**

**See Figure 6.**

Figure 6 c-h: I think it would be more informative to saturate the high end of the colorbar, so that the broader pattern of thickness increase is visible. As it stands, we only see what happens near the edges. This is subjective, but I feel like we are missing the broader picture.

**Following your suggestion, we have adjusted the colour bars of Figures 6 d-i.**

**See Figures 6 d-i.**

235: What is the climatological AMOC rate in this model (in piControl)?

**We have reported the climatological AMOC rate in "piControl" and "antwater 60S", as well as the difference between each experiment to show the change in the AMOC strength.**

**See Section 3.2; lines 275-276.**

230-260: It might be worth testing whether the AMOC/ACC timeseries are statistically different. As you say, there is a lot of variability.

**We have conducted and presented results from two-way student t-tests on the AMOC, AABW and ACC timeseries data that show that the results are statistically significance to the 99% confidence level.**

**See Section 3.2; line 277-278, Section 3.2; lines 285-286 and Section 3.2; lines 295-296.**

Figure 12: The changes in mid latitude precip are lost in the large changes near the equator. If you present the precip change as relative to the piControl (% as opposed to mm/day), it will take care of the climatological differences across latitudes, and perhaps be more meaningful (again, subjective).

**Following your suggestion, we have adjusted the colour bars of Figures 12 b-d to more clearly show climatological differences across latitudes.**

**See Figure 12.**

290: this explanation does not make physical sense to me, or I'm missing something. If injecting water around the coast makes it "able to spread horizontally across the surface of the SO before diffusing into the ocean depths", then injecting it even south of 60S would surely enable even more spread. Might a possible explanation might be that injection directly around Antarctica in antwater influences SO deep convection more directly than antwater60S, leading to more pronounced salinification at depth. I know Fig 18 shows similar AABW responses, but deep convection changes can be quite different to AABW (e.g. Chen et al., 2023)

**We have included a paragraph in the Sensitivity Study section that explains that restricting AIS meltwater to the coastline creates a freshwater cap at the ocean surface, preventing convection and inhibiting deep water formation, thereby inhibiting mixing and resulting in more pronounced salinification at depth. To confirm this, we have presented a map of annual mean mixed layer depth (MLD) from "antwater" and the difference in annual mean MLD between "antwater" and "antwater 60S"(Figure 15).**

**See Section 4; lines 335-340 and Figure 15.**

314: This "significant impact" statement seems to contradict the earlier results that AMOC changes are very small, and even AABW changes are not huge (but we do not know the relative changes). I would contend that the stratification mechanism that you mention next is likely more important.

**We have determined that the changes in AMOC and AABW are statistically significant to the 99% confidence level, suggesting that the global climate response to AIS meltwater is likely influenced by significant changes to the global overturning circulation.**

**See Section 3.2; line 277-278, Section 3.2; lines 285-286 and Section 3.2; lines 295-296.**

319: Sea-ice changes vs SAT changes are a bit chicken and egg. How much surface cooling could be explained by the ice-albedo feedback alone?

We have modified this section to reflect the dynamic relationship between sea ice change and SAT changes.
See Section 5; lines 370-372.

Other changes made:

1. We have rearranged the Methodology section to create a more consistent and approachable layout. As such, the changes in Section 2 appear substantial, but are predominantly just rearranging of existing paragraphs.

2. A minor error has been identified in the reporting of the change in AABW strength in the "antwater 60S" experiment. Instead of a 3.5 Sv change, this has been corrected to 2.5 Sv. As the results remain statistically significant and in line with results from previous studies, the original analysis and conclusions remain unchanged.

3. A minor error has been identified in Figure 18 whereby the caption did not include a description of plot 18a. This has been fixed.

4. We have renamed the final section to "Discussion and Conclusions" as it is quite discursive.